# CONTRASTIVE LEARNING OF MOLECULAR REPRESENTATION WITH FRAGMENTED VIEWS

## ABSTRACT

Molecular representation learning is a fundamental task for AI-based drug design and discovery. Contrastive learning is an attractive framework for this task, as also evidenced in various domains of representation learning, e.g., image, language, and speech. However, molecule-specific ways of constructing good positive or negative views in contrastive training under consideration of their chemical semantics have been relatively under-explored. In this paper, we consider a molecule as a bag of meaningful fragments, e.g., chemically informative substructures, by disconnecting a non-ring single bond as the semantic-preserving transformation. Then, we suggest to construct a complete (or incomplete) bag of fragments as the positive (or negative) views of a molecule: each fragment loses chemical substructures from the original molecule, while the union of the fragments does not. Namely, this provides easy positive and hard negative views simultaneously for contrastive representation learning so that it can selectively learn useful features and ignore nuisance features. Furthermore, we additionally suggest to optimize the torsional angle reconstruction loss around the fragmented bond to incorporate with 3D geometric structure in the pretraining dataset. Our experiments demonstrate that our scheme outperforms prior state-of-the-art molecular representation learning methods across various downstream molecule property prediction tasks.

## 1 INTRODUCTION

Obtaining discriminative representations of molecules is a long-standing research problem in chemistry (Morgan, 1965). Such a task is critical for many applications such as drug discovery (Capecchi et al., 2020) and material design (Gómez-Bombarelli et al., 2018), since it is a fundamental building block for various downstream tasks, e.g., molecule property prediction (Duvenaud et al., 2015) and molecule generation (Mahmood et al., 2021). Over the past decades, researchers have focused on handcrafting the molecular fingerprint representation which encodes the presence or absence of chemically meaningful substructures, e.g., functional groups, in a molecule (Rogers & Hahn, 2010).

Recently, graph neural networks (GNNs) (Kipf & Welling, 2016) have gained much attention as a framework to learn the molecular representation due to its remarkable performance for learning to predict chemical properties (Wu et al., 2018). However, they suffer from overfitting without much labeled training data (Rong et al., 2020b). To resolve this issue, researchers have investigated self-supervised learning to generate supervisory signals from a large amount of unlabeled molecules.

A notable approach on this line of work is contrastive learning, which learns a discriminative representation by maximizing the agreement of representations of "similar" positive views while minimizing the agreement of "dissimilar" negative views (Chen et al., 2020a); it has widely demonstrated its effectiveness for representation learning not only for molecules (Wang et al., 2021; 2022), but also for other domains, e.g., image (Chen et al., 2020a; He et al., 2019), video (Pan et al., 2021), language (Wu et al., 2020), and speech (Chung et al., 2021). Here, the common challenge for learning good representation is how to construct effective positive and negative views in a self-supervised manner.

For molecule contrastive representation learning, most prior works have utilized graph-augmentation techniques, e.g., edge/node drop, to produce positive views (You et al., 2020; 2021). However, such augmentations often fail to generate proper positive views of molecule graph, losing important chemical semantics from the anchor molecule, e.g., randomly inserting an edge of a graph may generate a non-realistic molecule (Fang et al., 2021b). Thus, *semantic-preserving transformation*

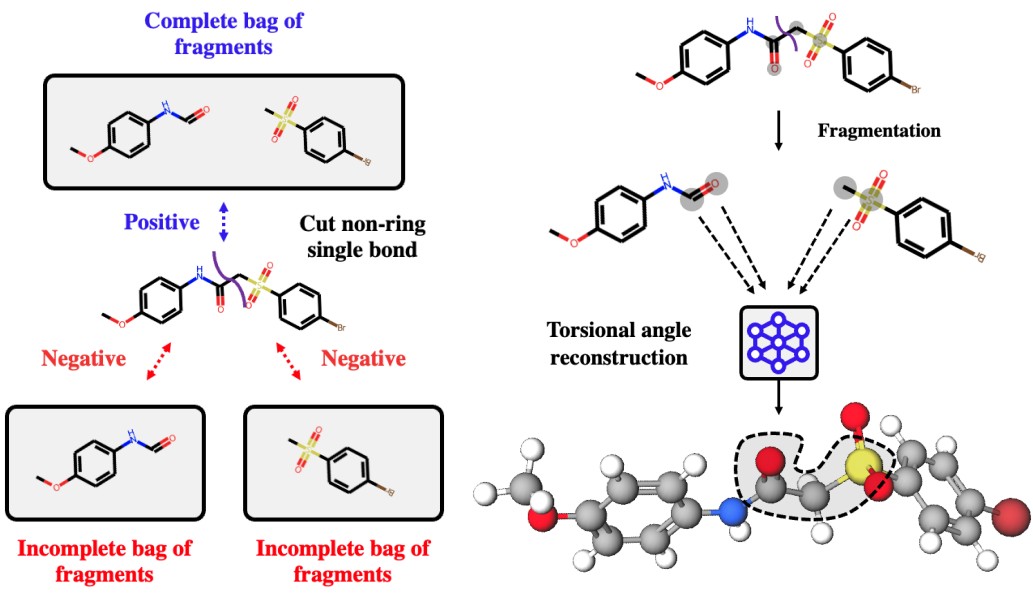

(a) Fragment-based contrastive learning         (b) Torsion reconstruction from fragments

Figure 1: Illustration of FragCL: contrastive learning of molecular representation with fragmented views. (a) Fragment-based view construction: We construct a bag of fragments from a molecule via fragmentation. A complete (or incomplete) bag of fragments is regarded as a positive (or negative) view of the original molecule. (b) Additionally, 3D contextual information can be learned by reconstructing the torsional angle around the fragmented bond.

should be carefully designed on molecule space to produce a "confident (or easy) positive" view of a molecule. On the other hand, despite the effectiveness of hard-to-discriminate negative samples has been widely evidenced in the contrastive representation learning literature of other domains (Robinson et al., 2020; Lee & Shin, 2022), such "hard negative" view construction is also not well-explored on molecule space yet.

**Contribution.** In this paper, we propose Fragment-based molecule Contrastive Learning (FragCL), a novel contrastive learning method using fragments to simultaneously generate easy positive and hard negative views of a molecule graph. FragCL consists of the following key ingredients with overall illustration provided in Figure 1.

- **Fragment-based positive view construction:** We construct a new positive view of molecules by decomposing it into a bag of meaningful fragments. We propose disconnecting a non-ring single bond of a molecule in half as the *semantic-preserving transformation*, since such a transformation preserves most of the chemically informative substructure, e.g., the number of heteroatoms and the existence of chemically informative substructures. Then, we suggest to regard a complete bag of resulting fragments as an easy positive view of a molecule.

- **Fragment-based negative view construction:** For negative views of a molecule, we consider (a) incomplete bag of fragments of its own and (b) complete bag of fragments of other molecules (in a mini-batch). Here, (a) is of strikingly different choice from prior molecule contrastive learning methods (You et al., 2020; Wang et al., 2021); existing works use the subgraphs of a molecule only as positive views, while we use an incomplete bag of fragments, which is a special kind of subgraph, as a negative view. Our intuition is that (a) becomes a hard-to-discriminate negative view as we fragment a molecule in half and incomplete bag of fragments, i.e., roughly half of important substructures of the anchor molecule are lost.

- **Torsional angle reconstruction from fragments:** We additionally propose a pretext task to incorporate 3D geometric context with fragments. We note that torsional angle, i.e., the angle between planes through two sets of three atoms having two atoms in common, defines several 3D contextual properties, e.g., the energy surface around the fragmented bond (Smith, 2008). Thus, the 2D graph encoder is able to learn meaningful 3D contextual information by reconstructing it from the fragments during training.

To demonstrate the effectiveness of FragCL, we extensively evaluate our method under various downstream molecule property prediction tasks on widely-used MoleculeNet (Wu et al., 2018) and ATOM3D (Townshend et al., 2020) benchmarks using 2D GNN pretrained by FragCL. Following Stärk et al. (2022) and Liu et al. (2022), we consider the most practical scenario: we pretrain with 2D and 3D paired unlabeled molecule dataset, and fine-tune with 2D downstream labeled molecule dataset.[1] We observe that FragCL outperforms prior state-of-the-art molecular representation learning method. For example, fragment-based molecule contrastive learning achieves the state-of-the-art performance in 7 out of 8 molecule property classification tasks on MoleculeNet, and all 7 molecule property regression tasks on ATOM3D.

## 2 PRELIMINARIES

### 2.1 GRAPH NEURAL NETWORKS FOR MOLECULES

In this work, we use graph neural networks (GNNs) to obtain discriminative representations of molecules. At a high-level, GNNs learn the molecular representation by applying (a) iterative neighborhood aggregation to acquire node representations from the molecule graph and (b) readout function to create a graph representation from the node representations.

**GNN for 2D molecule graphs.** A molecule can be represented as a 2D topological graph $\mathcal{G} = (\mathcal{V}, \mathcal{E})$ associated with a set of nodes (atoms) $\mathcal{V}$ and a set of edges (bonds) $\mathcal{E}$. Given a 2D graph $\mathcal{G}$, the $\ell$-th iteration of the neighborhood aggregation scheme is formulated as follows:

$$\widetilde{\mathbf{h}}_v^{(\ell)} \leftarrow \texttt{Aggregate}\big(\big\{(\mathbf{h}_u^{(\ell-1)}, \mathbf{h}_v^{(\ell-1)}, e_{uv}) : u \in \mathcal{N}(v)\big\}\big), \qquad (1)$$

$$\mathbf{h}_v^{(\ell)} \leftarrow \texttt{Combine}(\widetilde{\mathbf{h}}_v^{(\ell)}, \mathbf{h}_v^{(\ell-1)}). \qquad (2)$$

Here, $\mathcal{N}(v)$ is a set of adjacent nodes of $v$, and $\mathbf{h}_v^{(\ell)}$ is the $\ell$-th representation of node $v$ where $\mathbf{h}_v^{(0)}$ is the atom attribute associated with node $v$. Next, $e_{uv}$ is the bond attribute corresponding to the edge $\{u, v\} \in \mathcal{E}$. The $\texttt{Aggregate}$ and $\texttt{Combine}$ functions are GNN-specific components.

**GNN for 3D molecule graphs.** A molecule can be represented as a 3D geometric graph $\mathcal{G} = (\mathcal{V}, \mathcal{R})$ associated with a set of nodes (atoms) $\mathcal{V}$ and a set of 3D positions of atoms $\mathcal{R}$. Given a 3D graph $\mathcal{G}$, the $\ell$-th iteration of the neighborhood aggregation scheme is formulated as follows:

$$\widetilde{\mathbf{h}}_v^{(\ell)} \leftarrow \texttt{Aggregate}\big(\big\{(\mathbf{h}_u^{(\ell-1)}, \mathbf{h}_v^{(\ell-1)}, r_u, r_v) : u \in \mathcal{V}\big\}\big), \qquad (3)$$

$$\mathbf{h}_v^{(\ell)} \leftarrow \texttt{Combine}(\widetilde{\mathbf{h}}_v^{(\ell)}, \mathbf{h}_v^{(\ell-1)}). \qquad (4)$$

Here, $\mathbf{h}_v^{(\ell)}$ is the $\ell$-th representation of node $v$ where $\mathbf{h}_v^{(0)}$ is the atom attribute associated with node $v$. Next, $r_u$ and $r_v$ is the coordinate of $u$ and $v$, respectively. The $\texttt{Aggregate}$ and $\texttt{Combine}$ functions are GNN-specific components.

**Molecular representation extraction.** After $L$ times of the neighborhood aggregation, the 2D or 3D graph representation can be extracted from the final node representation $\mathbf{h}_v^{(L)}$ as follows:

$$f(\mathcal{G}) = \texttt{Readout}(\{\mathbf{h}_v^{(L)}\}_{v \in \mathcal{V}}),$$

where $\texttt{Readout}$ is a permutation invariant function such as node-wise mean-pooling.

### 2.2 CONTRASTIVE LEARNING

In general, contrastive learning aims to learn a meaningful data representation by minimizing its distance to *positive* data representations and maximizing distance to *negative* data representations. The key idea here is to incorporate the human prior in representation learning by carefully setting the positive and the negative relation between representations.

To this end, a number of contrastive learning objectives have been proposed, e.g., max-margin contrastive loss (Hadsell et al., 2006), triplet loss (Weinberger & Saul, 2009), or metric maximization

---

[1]Since obtaining an exact 3D geometric structure of a molecule is costly due to iterations of quantum calculations, it is highly likely that 3D information is not available in downstream tasks.

of local aggregation (Zhuang et al., 2019). The normalized temperature-scaled cross entropy loss (NT-Xent) (Chen et al., 2020a) is one of the widely used contrastive learning losses (Chen et al., 2020a;b; You et al., 2020). In this work, we follow GraphCL (You et al., 2020), which uses a variant of NT-Xent defined as follows:

$$\mathcal{L}_{\text{NT-Xent}}(\mathbf{z}, \mathbf{z}^+, \{\mathbf{z}^-\}; \tau) = -\log \frac{\exp(\text{sim}(\mathbf{z}, \mathbf{z}^+)/\tau)}{\sum_{\mathbf{z}^-} \exp(\text{sim}(\mathbf{z}, \mathbf{z}^-)/\tau)}, \tag{5}$$

where $\text{sim}(\mathbf{z}, \widetilde{\mathbf{z}}) = \mathbf{z}^\top \widetilde{\mathbf{z}}/\|\mathbf{z}\|_2\|\widetilde{\mathbf{z}}\|_2$ is the cosine similarity, $\tau$ is a temperature-scaling hyperparameter, $(\mathbf{z}, \mathbf{z}^+)$ and $(\mathbf{z}, \mathbf{z}^-)$ are positive and negative pairs of latent representations, respectively.

## 3 FRAGCL: FRAGMENTATION-BASED MOLECULE CONTRASTIVE LEARNING

In this work, we propose FragCL: **Frag**ment-based molecule **C**onstrastive **L**earning, a novel self-supervised learning framework for pretraining molecular representations. As mentioned in Section 1, we train a 2D GNN $f_{\text{2D}}(\cdot)$ as a molecular representation extractor by jointly training $f_{\text{3D}}(\cdot)$ on a pretraining dataset $\mathcal{D}$ containing both 2D and 3D molecule graphs.

Our key idea is to combine contrastive learning with *molecule fragmentation*: decomposing a molecule into a bag of meaningful fragments, e.g., chemically informative substructures. We construct a complete and incomplete bag of fragments as easy positive and hard negative views of a molecule, respectively. Additionally, we propose torsional angle reconstruction pretext task around the fragmented bond to further incorporate 3D geometric context in the pretraining dataset. The overall description of FragCL is illustrated in Figure 1. For each training step, FragCL performs the following operations:

1. **Molecule fragmentation:** Given a molecule $\mathcal{G} \in \mathcal{D}$, we obtain the 'bag of fragments' $\{\mathcal{G}', \mathcal{G}''\}$ by disconnecting a non-ring single bond whose absence is likely to preserve chemically informative substructures, i.e., $\mathcal{G}$ and $\{\mathcal{G}', \mathcal{G}''\}$ shares chemically informative substructures.

2. **Contrastive learning with bag of fragments:** We optimize the 2D GNN $f_{\text{2D}}(\cdot)$ and 3D GNN $f_{\text{3D}}(\cdot)$ along with the projection heads $g_{\text{2D}}(\cdot)$ and $g_{\text{3D}}(\cdot)$ via contrastive learning with our proposed positive/negative views, while maintaining the consistency of the representations from $f_{\text{2D}}(\cdot)$ and $f_{\text{3D}}(\cdot)$. To this end, we regard the complete (or incomplete) bag of fragments as a positive (or negative) view of each molecule graph (see Figure 1a). Additionally, we use the complete bag of fragments of other molecules in a mini-batch as negative views.

3. **Torsional angle reconstruction from fragments:** Given a bag of fragments $\{\mathcal{G}', \mathcal{G}''\}$ obtained by removing a non-ring single bond $(u, v)$ from $\mathcal{G}$ with $u \in \mathcal{G}'$ and $v \in \mathcal{G}''$, we reconstruct the torsional angle defined by a quartet of atoms $(s, u, v, t)$, linearly connected via covalent bonds in order in $\mathcal{G}$, to incorporate 3D contextual information of original molecule (see Figure 1b).

In the rest of this section, we provide details and rationale of our three components: molecule fragmentation in Section 3.1, contrastive learning with bag of fragments in Section 3.2, and torsional angle reconstruction in Section 3.3.

### 3.1 MOLECULE FRAGMENTATION

Our FragCL crucially relies on the molecule fragmentation to generate a bag of fragments from a molecule. While there exist various ways to break a molecule into disconnected components, we consider removal of a bond that (a) has the bond order of one and (b) is not a member of any ring. Note that double or higher-order bonds are capable of many reaction pathways such as nucleophilic attack and hydrogen addition (Smith, 2008), i.e., they are directly involved in chemically informative substructures. Thus, removing those bonds may alter the semantics of molecules seriously. On the other hand, single bonds does not determine the core semantic of the functional groups in general. For example, functional groups such as ketone, ester, and amide are categorized by the "carbonyl group", whose main functionalities are determined by C=O double bond. Also, the chemical functionalities of the ether and thioether come from the electronegative oxygen and sulfur atoms (not from the single bond), respectively (Smith, 2008). Thus, we choose only single bonds to fragment a molecule, and it is verified to be effective empirically in Section 4.3. Also, further discussion about the fragmentation strategy can be found in Appendix F.

The fragmentation of a 2D molecule $\mathcal{G}_{\text{2D}} = (\mathcal{V}, \mathcal{E})$ by a single non-ring bond $e \in \mathcal{E}$ produces *a complete bag of fragments* $\{\mathcal{G}'_{\text{2D}}, \mathcal{G}''_{\text{2D}}\}$ where $\mathcal{G}'_{\text{2D}}$ and $\mathcal{G}''_{\text{2D}}$ are components[2] in the graph induced on the edge set $\mathcal{E} \setminus \{e\}$. The edge $e \in \mathcal{E}$ is selected to minimize the difference of the number of atoms in $\mathcal{G}'_{\text{2D}}$ and $\mathcal{G}''_{\text{2D}}$, i.e., we split the original molecule in half. This is to prevent one of the fragments from containing almost all semantics of the original molecule because we consider an incomplete bag of fragments as a negative pair of the original molecule in Section 3.2.

The fragments on 3D molecule graph $\mathcal{G}_{\text{3D}}$ are defined by $\mathcal{G}'_{\text{3D}} = (\mathcal{V}', \mathcal{R}')$ and $\mathcal{G}''_{\text{3D}} = (\mathcal{V}'', \mathcal{R}'')$, where $\mathcal{V}'$ and $\mathcal{V}''$ are the sets of nodes of $\mathcal{G}'_{\text{2D}}$ and $\mathcal{G}''_{\text{2D}}$, and $\mathcal{R}'$ and $\mathcal{R}''$ are the corresponding coordinates of $\mathcal{V}'$ and $\mathcal{V}''$. When the context is clear, we drop the subscripts 2D and 3D.

## 3.2 CONTRASTIVE OBJECTIVE WITH BAG OF FRAGMENTS

We introduce a strategy to obtain positive and negative pairs of molecular representations and contrastive objectives based on the positive and negative pairs. In what follows, we describe the strategy to construct the pairs given a training batch $\{\mathcal{G}_i\}_{i=1}^{n}$ where $n$ denotes the batch size.

**Positive pairs.** The key insight is that our fragmentation preserves chemically meaningful substructures; for clarity, we denote $\mathcal{F}(\mathcal{G})$ as a set of all chemically informative substructures in $\mathcal{G}$. Then, each fragment $\mathcal{G}'$ and $\mathcal{G}''$, from our fragmentation scheme described in Section 3.1, satisfies $\mathcal{F}(\mathcal{G}_i) = \mathcal{F}(\mathcal{G}'_i) \cup \mathcal{F}(\mathcal{G}''_i)$ with a high probability. Based on this insight, we consider $(\mathcal{G}_i, \{\mathcal{G}'_i, \mathcal{G}''_i\})$ as a positive pair, i.e., positive pair consists of the original molecule and its complete bag of fragments. To facilitate the representation attraction of a molecule $\mathcal{G}_i$ and a bag of fragments $\{\mathcal{G}'_i, \mathcal{G}''_i\}$, we use a simple mixing strategy between $\mathbf{r}'_i := f(\mathcal{G}'_i)$ and $\mathbf{r}''_i := f(\mathcal{G}''_i)$ as $\mathbf{r}^{\text{mix}}_i := \frac{1}{n'+n''}(n' \times \mathbf{r}'_i + n'' \times \mathbf{r}''_i)$ with $n' := |\mathcal{G}'|$, and $n'' = |\mathcal{G}''|$; we attract $\mathbf{r}_i := f(\mathcal{G}_i)$ and $\mathbf{r}^{\text{mix}}_i$.

**Negative pairs.** For a given molecule $\mathcal{G}_i$, we construct negative pairs with (a) arbitrary complete bag of fragments $\{\mathcal{G}'_j, \mathcal{G}''_j\}$ with $i \neq j$, and (b) its own incomplete bag of fragments $\{\mathcal{G}'_i\}, \{\mathcal{G}''_i\}$. Intuitively, the negative pairs constructed using (a) is dissimilar to each other since an arbitrary complete bag of fragments $\{\mathcal{G}'_j, \mathcal{G}''_j\}$ has similar semantics with $\mathcal{G}_j$, which is distinct from the original molecule $\mathcal{G}_i$. Furthermore, we consider (b) as "hard" negative since deleting a large fragment significantly modifies the molecular semantics. Also, this pair is hard-to-discriminate since $\mathcal{G}'_i$ yields a common substructure to its original molecule $\mathcal{G}_i$.

Using positive and negative pairs, we train both 2D and 3D GNNs $f_{\text{2D}}(\cdot)$ and $f_{\text{3D}}(\cdot)$ along with projection heads $g_{\text{2D}}(\cdot)$ and $g_{\text{3D}}(\cdot)$ using following contrastive loss:

$$\mathcal{L}_{\text{2D,con}} := \frac{1}{n} \sum_{i=1}^{n} \mathcal{L}_{\text{NT-Xent}}(\mathbf{z}_{\text{2D},i}, \mathbf{z}^{\text{mix}}_{\text{2D},i}, \{\mathbf{z}^{\text{mix}}_{\text{2D},j}\}_{j \neq i} \cup \{\mathbf{z}'_{\text{2D},i}, \mathbf{z}''_{\text{2D},i}\}; \tau), \tag{6}$$

$$\mathcal{L}_{\text{3D,con}} := \frac{1}{n} \sum_{i=1}^{n} \mathcal{L}_{\text{NT-Xent}}(\mathbf{z}_{\text{3D},i}, \mathbf{z}^{\text{mix}}_{\text{3D},i}, \{\mathbf{z}^{\text{mix}}_{\text{3D},j}\}_{j \neq i} \cup \{\mathbf{z}'_{\text{3D},i}, \mathbf{z}''_{\text{3D},i}\}; \tau), \tag{7}$$

where $\mathbf{z}_i$, $\mathbf{z}'_i$, $\mathbf{z}''_i$, and $\mathbf{z}^{\text{mix}}_i$ denote latent representations projected by $g(\cdot)$ from $\mathbf{r}_i$, $\mathbf{r}'_i$, $\mathbf{r}''_i$, and $\mathbf{r}^{\text{mix}}_i$, respectively (Chen et al., 2020a; You et al., 2020). Namely, we plug in our proposed positive and negative samples into the NT-Xent loss of equation 5.

To facilitate the representation learning on the multi-modality of 2D and 3D graphs of molecules, we consider each 2D and 3D view of a molecule as a positive pair, following the common practice to deal with the multi-modal representation learning (Radford et al., 2021; Stärk et al., 2022; Liu et al., 2022):

$$\mathcal{L}_{\{\text{2D,3D}\},\text{con}} := \frac{1}{2n} \sum_{i=1}^{n} \left( \mathcal{L}_{\text{NT-Xent}}(\mathbf{z}_{\text{2D},i}, \mathbf{z}_{\text{3D},i}, \{\mathbf{z}_{\text{3D},j}\}_{j=1}^{n}; \tau) + \mathcal{L}_{\text{NT-Xent}}(\mathbf{z}_{\text{3D},i}, \mathbf{z}_{\text{2D},i}, \{\mathbf{z}_{\text{2D},j}\}_{j=1}^{n}; \tau) \right),$$

$$\tag{8}$$

Finally, we obtain our contrastive objective combining above:

$$\mathcal{L}_{\text{con}} := \mathcal{L}_{\text{2D,con}} + \mathcal{L}_{\text{3D,con}} + \mathcal{L}_{\{\text{2D,3D}\},\text{con}}. \tag{9}$$

After pretraining with the loss using (unlabeled) molecule dataset, we transfer pretrained 2D GNN $f_{\text{2D}}(\cdot)$ for fine-tuning to a downstream task; note that $f_{\text{3D}}(\cdot)$, $g_{\text{2D}}(\cdot)$, and $g_{\text{3D}}(\cdot)$ are not transferred.

---

[2]A component of a graph is a connected subgraph that is not part of any larger connected subgraph.

### 3.3 TORSIONAL ANGLE RECONSTRUCTION FROM FRAGMENTS

We propose an additional pretext task to incorporate the 3D contextual information into our 2D GNN $f_{\text{2D}}(\cdot)$. To this end, $f_{\text{2D}}(\cdot)$ learns to reconstruct the torsional angle around the fragmented bond from fragments of a 2D molecule graph $\mathcal{G}'_{\text{2D}}$ and $\mathcal{G}''_{\text{2D}}$.

Our reconstruction target, torsional angle, is defined by a quartet of atoms. Let $(s, u, v, t)$ be a tuple of vertices (atoms) in $\mathcal{G}$, which is linearly connected via covalent bonds in order. The torsional angle is defined by the angle between the planes defined by $(s, u, v)$ and $(u, v, t)$, and it encodes important 3D contextual properties, e.g., energy surface around the atoms (Smith, 2008). Thus, by the reconstruction of the torsional angle from fragments $\{\mathcal{G}', \mathcal{G}''\}$ obtained by the removal of $(u, v)$ from $\mathcal{G}$, the 2D GNN $f_{\text{2D}}(\cdot)$ would learn the 3D contextual properties around the fragmented bond. The loss function is defined as follows:

$$\mathbf{z}_{\text{rot},i}(s, u, v, t) := g_{\text{rot}}([\mathbf{h}_{\text{2D},s,i}; \mathbf{h}_{\text{2D},u,i}; \mathbf{h}_{\text{2D},v,i}; \mathbf{h}_{\text{2D},t,i}]) \tag{10}$$

$$\mathbf{z}_{\text{abs},i}(s, u, v, t) := g_{\text{abs}}([\mathbf{h}_{\text{2D},s,i}; \mathbf{h}_{\text{2D},u,i}; \mathbf{h}_{\text{2D},v,i}; \mathbf{h}_{\text{2D},t,i}]) \tag{11}$$

$$\mathcal{L}_{\text{torsion}} := \frac{1}{n} \sum_{i=1}^{n} \big( \mathcal{L}_{\text{BCE}}(\mathbf{z}_{\text{rot},i}(s, u, v, t), y_{\text{rot},i}) + \mathcal{L}_{\text{CE}}(\mathbf{z}_{\text{abs},i}(s, u, v, t), y_{\text{abs},i}) \tag{12}$$

$$+ \mathcal{L}_{\text{BCE}}(\mathbf{z}_{\text{rot},i}(t, v, u, s), y_{\text{rot},i}) + \mathcal{L}_{\text{CE}}(\mathbf{z}_{\text{abs},i}(t, v, u, s), y_{\text{abs},i}) \big), \tag{13}$$

where $\mathbf{h}_{\text{2D},\{s,u,v,t\},i}$ is the representation of $\{s, u, v, t\}$'th node from $f_{\text{2D}}(\cdot)$, respectively. And $y_{\text{rot},i}$ is a binary label for the rotation direction of torsional angle, i.e., clockwise or counter-clockwise, and $y_{\text{abs},i}$ is a binned label for the absolute torsional angle value between $0$ and $\pi$ of $i$-th ground state molecule. Also, $g_{\text{rot}}$ and $g_{\text{abs}}$ are projection functions, and $\mathcal{L}_{\text{BCE}}$ and $\mathcal{L}_{\text{CE}}$ are binary cross-entropy and cross-entropy loss function, respectively. Since the torsional angle defined by $(s, u, v, t)$ and $(t, v, u, s)$ are the same, we learn to reconstruct the torsional angle from both tuples.

### 3.4 OVERALL TRAINING OBJECTIVE

From the discussion of Section 3.2 and 3.3, we finally propose our training loss function. In summary, we set (a) the complete bag of fragments as a positive view of molecule, (b) an incomplete bag of its own and the complete bag of fragments of other molecules as negative views of molecule while maximizing the consistency of the outputs of 2D and 3D GNNs. Additionally, we incorporate 3D contextual information by reconstructing the torsional angle around the fragmented bond. The specific loss function is as follows:

$$\mathcal{L}_{\text{FragCL}} := \mathcal{L}_{\text{con}} + \alpha \mathcal{L}_{\text{torsion}}, \tag{14}$$

where $\alpha$ is a hyperparameter to control the contribution of the torsion reconstruction objective, and we simply use $\alpha = 1$ for all experiments. The ablation study on $\alpha$ can be found in Appendix E.

## 4 EXPERIMENTS

In this section, we evaluate the effectiveness of our FragCL. To this end, we extensively compare FragCL with the existing molecular graph representation learning methods. We evaluate FragCL and baselines on various downstream molecule property prediction tasks after pretraining on (unlabeled) molecule dataset. Also, we perform the extensive ablation study on FragCL to confirm that each of our components plays an important role for effectively discriminating the molecules.

### 4.1 EXPERIMENTAL SETUP

In what follows, we describe our experimental setup, where we largely follow prior works (Hu et al., 2019; Liu et al., 2022). Detailed information can be found in Appendix A.

**Datasets.** For self-supervised pretraining, we use a 50k unlabeled molecule dataset with 2D and 3D paired molecule graphs from GEOM (Axelrod & Gomez-Bombarelli, 2022). After pretraining, transfer learning is performed on (a) binary classification tasks from MoleculeNet benchmark (Wu

Table 1: Comparison of test ROC-AUC score on MoleculeNet downstream property classification benchmarks. We report mean and standard deviation over 3 different seeds. We mark the best mean score bold. Additionally, we mark the average scores within one standard deviation from the highest average score to be bold. We remark by (*) when we use the score reported by Liu et al. (2022). Otherwise, we reproduced scores under the same setup.

| Methods | BBBP | Tox21 | ToxCast | Sider | Clintox | MUV | HIV | Bace |
|---|---|---|---|---|---|---|---|---|
| - | $65.4_{\pm2.4}$ | $74.9_{\pm0.8}$ | $61.6_{\pm1.2}$ | $58.0_{\pm2.4}$ | $58.8_{\pm5.5}$ | $71.0_{\pm2.5}$ | $75.3_{\pm0.5}$ | $72.6_{\pm4.9}$ |
| Pretrained on 2D molecule graph | | | | | | | | |
| EdgePred* | $64.5_{\pm3.1}$ | $74.5_{\pm0.4}$ | $60.8_{\pm0.5}$ | $56.7_{\pm0.1}$ | $55.8_{\pm6.2}$ | $73.3_{\pm1.6}$ | $75.1_{\pm0.8}$ | $64.6_{\pm4.7}$ |
| AttrMask* | $70.2_{\pm0.5}$ | $74.2_{\pm0.8}$ | $62.5_{\pm0.4}$ | $60.4_{\pm0.6}$ | $68.6_{\pm9.6}$ | $73.9_{\pm1.3}$ | $74.3_{\pm1.3}$ | $77.2_{\pm1.4}$ |
| GPT-GNN* | $64.5_{\pm1.1}$ | $75.3_{\pm0.5}$ | $62.2_{\pm0.1}$ | $57.5_{\pm4.2}$ | $57.8_{\pm3.1}$ | $76.1_{\pm2.3}$ | $75.1_{\pm0.2}$ | $77.6_{\pm0.5}$ |
| Infomax* | $69.2_{\pm0.8}$ | $73.0_{\pm0.7}$ | $62.0_{\pm0.3}$ | $59.2_{\pm0.2}$ | $75.1_{\pm5.0}$ | $74.0_{\pm1.5}$ | $74.5_{\pm1.8}$ | $73.9_{\pm2.5}$ |
| ContextPred* | $\mathbf{71.2}_{\pm0.9}$ | $73.3_{\pm0.5}$ | $62.8_{\pm0.3}$ | $59.3_{\pm1.4}$ | $73.7_{\pm4.0}$ | $72.5_{\pm2.2}$ | $75.8_{\pm1.1}$ | $78.6_{\pm1.4}$ |
| GraphLoG* | $67.8_{\pm1.7}$ | $73.0_{\pm0.3}$ | $62.2_{\pm0.4}$ | $57.4_{\pm2.3}$ | $62.0_{\pm1.8}$ | $73.1_{\pm1.7}$ | $73.4_{\pm0.6}$ | $78.8_{\pm0.7}$ |
| G-Contextual* | $70.3_{\pm1.6}$ | $75.2_{\pm0.3}$ | $62.6_{\pm0.3}$ | $58.4_{\pm0.6}$ | $59.9_{\pm8.2}$ | $72.3_{\pm0.9}$ | $75.9_{\pm0.9}$ | $79.2_{\pm0.3}$ |
| G-Motif* | $66.4_{\pm3.4}$ | $73.2_{\pm0.8}$ | $62.6_{\pm0.5}$ | $60.6_{\pm1.1}$ | $77.8_{\pm2.0}$ | $73.3_{\pm2.0}$ | $73.8_{\pm1.4}$ | $73.4_{\pm4.0}$ |
| GraphCL* | $67.5_{\pm3.3}$ | $75.0_{\pm0.3}$ | $62.8_{\pm0.2}$ | $60.1_{\pm1.3}$ | $78.9_{\pm4.2}$ | $\mathbf{77.1}_{\pm1.0}$ | $75.0_{\pm0.4}$ | $68.7_{\pm7.8}$ |
| JOAO* | $66.0_{\pm0.6}$ | $74.4_{\pm0.7}$ | $62.7_{\pm0.6}$ | $60.7_{\pm1.0}$ | $66.3_{\pm3.9}$ | $\mathbf{77.0}_{\pm2.2}$ | $76.6_{\pm0.5}$ | $72.9_{\pm2.0}$ |
| JOAOv2 | $67.2_{\pm3.6}$ | $75.0_{\pm0.7}$ | $63.5_{\pm0.3}$ | $60.6_{\pm0.4}$ | $77.1_{\pm3.9}$ | $73.4_{\pm3.4}$ | $\mathbf{77.7}_{\pm1.1}$ | $71.7_{\pm0.5}$ |
| MGSSL | $67.3_{\pm0.9}$ | $74.5_{\pm0.2}$ | $63.6_{\pm0.4}$ | $58.4_{\pm0.2}$ | $75.4_{\pm3.8}$ | $73.9_{\pm1.4}$ | $77.2_{\pm2.5}$ | $76.2_{\pm1.3}$ |
| MolCLR | $67.6_{\pm0.6}$ | $74.4_{\pm1.3}$ | $62.9_{\pm0.2}$ | $58.7_{\pm1.1}$ | $57.9_{\pm3.0}$ | $70.8_{\pm2.8}$ | $75.4_{\pm1.2}$ | $74.6_{\pm3.5}$ |
| 2D-FragCL (Ours) | $68.2_{\pm3.0}$ | $74.7_{\pm0.3}$ | $62.4_{\pm0.7}$ | $60.3_{\pm0.6}$ | $82.7_{\pm1.3}$ | $\mathbf{77.9}_{\pm1.0}$ | $76.1_{\pm1.4}$ | $77.2_{\pm1.0}$ |
| Pretrained on 2D and 3D molecule graph | | | | | | | | |
| GraphMVP* | $68.5_{\pm0.2}$ | $74.5_{\pm0.4}$ | $62.7_{\pm0.1}$ | $62.3_{\pm1.6}$ | $79.0_{\pm2.5}$ | $75.0_{\pm1.4}$ | $74.8_{\pm1.4}$ | $76.8_{\pm1.1}$ |
| GraphMVP-G* | $\mathbf{70.8}_{\pm0.5}$ | $75.9_{\pm0.5}$ | $63.1_{\pm0.2}$ | $60.2_{\pm1.1}$ | $79.1_{\pm2.8}$ | $\mathbf{77.7}_{\pm0.6}$ | $76.0_{\pm0.1}$ | $79.3_{\pm1.5}$ |
| GraphMVP-C* | $\mathbf{72.4}_{\pm1.6}$ | $74.4_{\pm0.2}$ | $63.1_{\pm0.4}$ | $\mathbf{63.9}_{\pm1.2}$ | $77.5_{\pm4.2}$ | $75.0_{\pm1.0}$ | $77.0_{\pm1.2}$ | $\mathbf{81.2}_{\pm0.9}$ |
| 3D-InfoMax | $67.9_{\pm1.2}$ | $75.3_{\pm0.3}$ | $\mathbf{64.6}_{\pm0.4}$ | $59.6_{\pm0.7}$ | $\mathbf{89.7}_{\pm0.5}$ | $76.7_{\pm0.6}$ | $73.4_{\pm1.2}$ | $79.9_{\pm0.9}$ |
| FragCL (Ours) | $\mathbf{70.9}_{\pm1.6}$ | $\mathbf{76.2}_{\pm0.2}$ | $\mathbf{64.2}_{\pm0.5}$ | $61.9_{\pm0.9}$ | $\mathbf{89.9}_{\pm1.2}$ | $\mathbf{77.8}_{\pm0.6}$ | $\mathbf{77.8}_{\pm0.5}$ | $80.9_{\pm1.0}$ |

et al., 2018), and (b) regression tasks from ATOM3D benchmark (Townshend et al., 2020). Detailed explanations for each downstream task can be found in Appendix D.

**Baselines.** We follow the baselines considered in GraphMVP (Liu et al., 2022), EdgePred (Hamilton et al., 2017), AttrMask (Hu et al., 2019), GPT-GNN (Hu et al., 2020), Infomax (Sun et al., 2019), ContextPred (Hu et al., 2019), GraphLoG (Xu et al., 2021), G-{Contextual, Motif} (Rong et al., 2020a), GraphCL (You et al., 2020), and JOAO (You et al., 2021). Additionally, we compare with newly-proposed 3D-Infomax (Stärk et al., 2022). GraphMVP, GraphMVP-{G,C}, and 3D-Infomax utilize both 2D and 3D molecule graphs in pretraining, while other baselines are based on 2D molecule graph. Detailed information of baselines can be found in Appendix A. For more comparison, we add 2D-FragCL, which is pretrained on only 2D molecule graphs with $\mathcal{L}_{2D,con}$ in equation 6. More comparison on 2D-FragCL and 2D molecule graph representation learning methods on ZINC15 (Sterling & Irwin, 2015) pretraining dataset can be found in Appendix B.

**Experimental details.** We consider 5-layer graph isomorphism network (GIN) (Xu et al., 2019) and 6-layer SchNet (Schütt et al., 2017) with mean pooling as readout function of our 2D GNN $f_{2D}(\cdot)$ and 3D GNN $f_{3D}(\cdot)$. The configuration is drawn from GraphMVP (Liu et al., 2022) for a fair comparison. Note that we only transfer $f_{2D}(\cdot)$ to fine-tune on 2D molecule graphs in the downstream tasks. Further details can be found in Appendix A.

## 4.2 MAIN RESULTS

In our experiments, we achieve state-of-the-art performance in downstream molecule property prediction tasks. In both MoleculeNet and ATOM3D downstream tasks, FragCL consistently achieves the state-of-the-art ROC-AUC and MAE score, respectively.

**MoleculeNet classification task.** As reported in Table 1, FragCL achieves the best average test ROC-AUC score when transferred to MoleculeNet (Wu et al., 2018) downstream tasks. Specifically, our method achieves the state-of-the-art performance on 7 out of 8 downstream tasks. We emphasize

Table 2: Comparison of test MAE score on ATOM3D downstream quantum property regression benchmarks. We report mean and standard deviation over 3 different seeds. We mark the best mean score bold. Additionally, we mark the average scores within one standard deviation from the highest average score to be bold.

| Methods | ZPVE | $\mu$ | $\alpha$ | $C_v$ | LUMO | HOMO | $\varepsilon_{gap}$ | $R^2$ | $U_0$ | $U_{298}$ | $H_{298}$ | $G_{298}$ |
|---|---|---|---|---|---|---|---|---|---|---|---|---|
| - | $49.7_{\pm8.7}$ | $0.428_{\pm0.002}$ | $0.666_{\pm0.060}$ | $0.255_{\pm0.008}$ | $84.8_{\pm0.7}$ | $85.6_{\pm1.2}$ | $124_{\pm1}$ | $28.8_{\pm0.9}$ | $74.9_{\pm9.5}$ | $68.3_{\pm11.2}$ | $72.0_{\pm10.6}$ | $71.2_{\pm2.9}$ |
| CP | $30.7_{\pm2.1}$ | $0.416_{\pm0.002}$ | $0.633_{\pm0.032}$ | $0.219_{\pm0.005}$ | $85.6_{\pm0.9}$ | $86.7_{\pm0.9}$ | $124_{\pm2}$ | $25.4_{\pm0.3}$ | $60.0_{\pm6.1}$ | $60.8_{\pm10.1}$ | $65.2_{\pm11.1}$ | $60.8_{\pm10.1}$ |
| GraphCL | $27.2_{\pm0.5}$ | $0.419_{\pm0.005}$ | $0.589_{\pm0.039}$ | $0.225_{\pm0.002}$ | $85.0_{\pm0.5}$ | $85.3_{\pm0.3}$ | $121_{\pm1}$ | $25.1_{\pm0.4}$ | $57.2_{\pm4.2}$ | $58.9_{\pm3.9}$ | $55.1_{\pm2.7}$ | $59.9_{\pm4.5}$ |
| GraphMVP-C | $22.7_{\pm1.8}$ | $0.423_{\pm0.001}$ | $0.521_{\pm0.038}$ | $\mathbf{0.199}_{\pm0.008}$ | $85.2_{\pm0.6}$ | $86.5_{\pm0.3}$ | $124_{\pm1}$ | $\mathbf{25.2}_{\pm0.1}$ | $\mathbf{37.4}_{\pm2.0}$ | $\mathbf{38.6}_{\pm3.6}$ | $\mathbf{39.2}_{\pm4.0}$ | $43.6_{\pm2.2}$ |
| 3D-Infomax | $\mathbf{22.2}_{\pm1.6}$ | $0.412_{\pm0.004}$ | $\mathbf{0.492}_{\pm0.006}$ | $0.202_{\pm0.004}$ | $84.9_{\pm1.0}$ | $\mathbf{83.2}_{\pm0.9}$ | $121_{\pm0}$ | $24.8_{\pm0.3}$ | $40.3_{\pm1.3}$ | $\mathbf{39.7}_{\pm1.2}$ | $\mathbf{38.7}_{\pm2.8}$ | $\mathbf{38.0}_{\pm1.3}$ |
| FragCL (Ours) | $23.4_{\pm1.2}$ | $\mathbf{0.409}_{\pm0.001}$ | $\mathbf{0.488}_{\pm0.023}$ | $0.201_{\pm0.007}$ | $\mathbf{80.9}_{\pm1.3}$ | $\mathbf{83.0}_{\pm0.6}$ | $\mathbf{118}_{\pm1}$ | $\mathbf{24.4}_{\pm0.8}$ | $\mathbf{39.1}_{\pm3.0}$ | $\mathbf{40.0}_{\pm3.6}$ | $\mathbf{40.9}_{\pm2.9}$ | $\mathbf{37.7}_{\pm2.2}$ |

that the improvement of FragCL is consistent over downstream tasks. For example, 3D-InfoMax (Liu et al., 2022) achieves the best performance on ToxCast, while it fails to generalize on Sider, resulting in even lower ROC-AUC score compared to several baselines pretrained on 2D dataset. On the other hand, FragCL shows the best average performance with no such failure case, i.e., FragCL learns well-generalizable representations over several downstream tasks. Also, 2D-FragCL even performs better than GraphMVP (pretrained on both 2D and 3D dataset) on 4 out of 8 tasks. This verifies the effectiveness of our proposed view construction strategy between a molecule and its bag of fragments.

**ATOM3D regression task.** Table 2 shows the overall results of transfer learning on ATOM3D (Townshend et al., 2020) molecule property regression benchmark. Overall, our FragCL outperforms the baselines, achieving the state-of-the-art performance over all considered downstream tasks of ATOM3D. We emphasize that our FragCL shows the best performance when fine-tuned on both MoleculeNet and ATOM3D downstream dataset, which verifies that FragCL learns representations generally applicable over distinct fine-tuning data distributions.

## 4.3 ABLATION STUDY

**Fragment-based view construction.** To recognize the effectiveness of our view construction strategy, we conduct an ablation study on regarding the complete (incomplete) bag of fragments as a positive (negative) view in Table 3. We start our ablation by considering an incomplete bag of fragments, i.e., a subgraph, as a positive view of the original molecule, which is similar to the prior methods (You et al., 2020; 2021; Wang et al., 2021). The improvement $70.4 \rightarrow 71.7 \rightarrow 72.1$ is from our careful easy positive and hard negative view construction strategy via molecule fragmentation from its own molecule, which is different from existing molecular contrastive representation learning methods. Also, regarding the incomplete bag of fragments as a neutral view, i.e., complete bag of fragments of other molecules as only negatives, greatly improves its non-pretraining counterpart by $67.2 \rightarrow 71.9$ and it verifies the effectiveness of our negative view construction in a mini-batch.

**Strategy to construct fragments.** To verify the effectiveness of our bond-disconnecting strategy to obtain the bag of fragments, we conduct experiments with an alternative strategy: cutting random non-ring bonds, i.e., including single, double and triple bond. As shown in Table 3, our single-bond disconnecting strategy improves disconnecting a random non-ring bond by $72.1 \rightarrow 72.4$ for $\mathcal{L}_{2D,con}$ and $73.3 \rightarrow 74.0$ for $\mathcal{L}_{con}$. Note that the margin is larger in $\mathcal{L}_{con}$ since $\mathcal{L}_{con}$ utilizes both 2D and 3D fragments, while $\mathcal{L}_{2D,con}$ is based on only 2D fragments. The results verify our *semantic-preserving transformation*, i.e., disconnecting a non-ring single bond, successfully creates an easy positive view by maintaining informative substructures of molecules.

**The proposed loss.** Our loss design $\mathcal{L}_{FragCL}$ (equation 14) combines several components proposed in Section 3, and here we validate that each of the components has an individual effect in improving the performance. Table 3 shows that 2D-FragCL, trained with $\mathcal{L}_{2D,con}$ as its loss function, effectively learns discriminative representations from fragmented 2D views by obtaining $67.2 \rightarrow 72.4$ from its non-pretraining counterpart. Also, jointly training 3D GNN with 2D GNN improves 2D-FragCL by $72.4 \rightarrow 74.0$, and incorporating 2D GNN with 3D torsional angle further boosts the performance by $74.0 \rightarrow 75.0$. This confirms that each component of our FragCL plays its role effectively. Also, the effectiveness of the torsional angle reconstruction task is further verified in the advanced torsion-aware architecture SphereNet (Liu et al., 2021), by improving the average performance by $74.4 \rightarrow 75.1$. The detailed results with SphereNet can be found in Appendix G.

Table 3: Ablation of FragCL with the average ROC-AUC score of 8 downstream tasks on MoleculeNet. Detailed results can be found in Appendix E.

| Complete bag of fragments | Incomplete bag of fragments | Disconnecting bond | Pretraining loss | Avg |
|---|---|---|---|---|
| - | - | - | - | 67.2 |
| Neutral | Positive | Random | $\mathcal{L}_{\text{2D,con}}$ | 70.4 |
| Positive | Positive | Random | $\mathcal{L}_{\text{2D,con}}$ | 71.7 |
| Positive | Negative | Random | $\mathcal{L}_{\text{2D,con}}$ | 72.1 |
| Positive | Negative | Single | $\mathcal{L}_{\text{2D,con}}$ | 72.4 |
| Positive | Neutral | Single | $\mathcal{L}_{\text{2D,con}}$ | 71.9 |
| Positive | Negative | Single | $\mathcal{L}_{\text{2D,con}}$ | 72.4 |
| Positive | Negative | Random | $\mathcal{L}_{\text{con}}$ | 73.3 |
| Positive | Negative | Single | $\mathcal{L}_{\text{con}}$ | 74.0 |
| Positive | Negative | Single | $\mathcal{L}_{\text{FragCL}}$ | 75.0 |

## 5 RELATED WORKS

**Molecular representation learning.** Researchers have paid attention to learning molecular representations for downstream tasks given a massive unlabeled molecule dataset. *self-supervised molecular representation learning*. One of the self-supervised learning techniques for molecular representation learning is generative learning. For example, those methods reconstruct the corrupted input as pre-defined pretext tasks (Hamilton et al., 2017; Hu et al., 2019; Rong et al., 2020a; Zhang et al., 2021). Another large portion of graph self-supervised learning consists of contrastive learning, which aims to learn a representation by pulling positive samples together and pushing negative samples apart. Zhu et al. (2021a) considers SMILES 1D string and 2D molecule of a molecule as a positive pair. You et al. (2020; 2021); Wang et al. (2021); Zhang et al. (2020) utilize augmentation schemes to produce positive view of molecule graphs, Fang et al. (2021a); Sun et al. (2021); Wang et al. (2022) mitigate the effect of semantically similar molecules in the negative samples (Zhu et al., 2021b). Recently, there are pioneering attempts to incorporate 2D topology of molecule with paired 3D geometric structures (Stärk et al., 2022; Liu et al., 2022).

**Molecule fragmentation.** Recently, several machine learning researches for molecule generation (Maziarz et al., 2021; Jin et al., 2018; 2020) regard a molecule as a combination of semantically important pieces, e.g., fragment. Such approach is natural in chemical sense, since the property of a molecule is largely determined by its chemically important substructures; not by atom-level features (Smith, 2008). Zhang et al. (2021) has adapted this concept to molecular representation learning by considering the reconstruction of substructures as a generative pretext task. However, there are remaining challenges: since the dictionary of substructure grows as the size of pretraining dataset increases, the reconstruction is infeasible in large scale pretraining dataset.

On the other hand, some research on contrastive molecular representation learning has utilized a substructure of molecule in their training objective. For example, You et al. (2020); Wang et al. (2021); Zhang et al. (2020) construct a positive view of a molecule as its substructure. These works differ from ours such that they focus on maximizing similarity between representations of a molecule and its single fragment, i.e., positive view, while FragCL considers a single fragment as the (hard) *negative* view. Such an opposite strategy is possible for FragCL as it instead considers a complete bag of fragments as the (easy) positive view. In addition, Wang et al. (2022) contrasts the representations of molecule-molecule and fragment-fragment pairs, while FragCL considers molecule-fragment pairs.

## 6 CONCLUSION

We present FragCL, a new contrastive molecular representation learning method based on chemical prior: A molecule can be viewed as a "bag of fragments", which consists of chemically meaningful structure. Based on this insight, we regard the complete bag of fragment as an easy positive view of a given molecule. Moreover, we set an incomplete bag of fragments to be a hard negative view of its own molecule since they differ in chemical property, while sharing some chemical substructures. Additionally, FragCL learns 3D contextual information by reconstructing the torsional angle around the fragmented bond. Experiments show that FragCL outperforms in predicting molecule property thanks to our pretraining strategy guided by clever fragment-based views of molecule graphs.

ETHICS STATEMENT

This work will facilitate research in molecular representation learning, which can speed up the processing of many important downstream tasks such as molecule property prediction and molecule generation. However, malicious use of well-learned molecular expressions poses a potential threat of creating hazardous substances, such as toxic chemical substances or biological weapons. On the other hand, molecule representation is also essential for creating defense mechanisms against harmful substances, so the careful use of our work, FragCL, can lead to more positive effects.

REPRODUCIBILITY STATEMENT

We describe the implementation details for self-supervised pretraining and fine-tuning in Appendix A, and provide our source code in the supplementary material. For 2D-FragCL, we use a single NVIDIA GeForce RTX 2080 Ti GPU and 40 CPU cores (Intel(R) Xeon(R) CPU E5-2630 v4 @ 2.20GHz), and we use a single NVIDIA-A100-SXM GPU and 124 CPU cores (AMD EPYC 7542 32-Core Processor) for FragCL.

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

# A    EXPERIMENTAL DETAILS

**Training details.** We follow the training setup considered in GraphMVP (Liu et al., 2022): Specifically, we use a batch size of 256 and no weight decay. Also, we set the temperature $\tau$ in equation 5 as 0.1 for overall experiments. We use {Nodedrop, Attrmask, identity} randomly, i.e., $\frac{1}{3}$ probability for each fragment and the original 2D molecule graphs, and Gaussian noise $\mathcal{N}(0, I)$ to each coordinate of 3D molecule graphs. When pretraining, we excluded about 0.1% of molecules that does not satisfy our fragmentation scheme (51 out of 50k molecules in GEOM 50k dataset. When Nodedrop or Attrmask is used, we drop/mask the portion of 0.1 vertices from the total vertices. For self-supervised pretraining, we train for 100 epochs using Adam optimizer (Kingma & Ba, 2014) with a learning rate of 0.001 and no dropout. Our code is based on open-source codes of GraphMVP[3] and GraphCL[4].

For 2D-FragCL and other reproduced 2D baselines, we exclude implicit hydrogens in molecule graph, following the common frameworks of (You et al., 2020; 2021). For FragCL and 3D-InfoMax, we include implicit hydrogens into molecule graph, following (Liu et al., 2022) that utilizes the 3D coordinates of hydrogen atoms provided in GEOM dataset (Axelrod & Gomez-Bombarelli, 2022). We exclude an incomplete bag of fragments $\mathcal{G}'$ from the negative pair set of the anchor molecule $\mathcal{G}$ when $|\mathcal{G}'| \geq 0.7 \times |\mathcal{G}|$, to prevent the incomplete bag of fragments $\mathcal{G}'$ from becoming a semantically positive view. For torsional angle prediction task, we construct the quartet of atoms $(s, u, v, t)$ for the fragmented bond $(u, v)$ so that $s, t$ are non-hydrogen atoms, and the binning of $y_{\text{abs}}$ splits 0 to $\pi$ into 9 uniform bins. We use 2-layer and 3-layer MLP for $g_{rot}(\cdot)$ and $g_{abs}(\cdot)$, respectively. Also, $g_{rot}(\cdot)$ and $g_{abs}(\cdot)$ shares the first layer of MLP.

**Evaluation on MoleculeNet downstream tasks.** Following the baselines, we use *scaffold split* (Chen et al., 2012), which splits the molecules based on their substructures. We use the split ratio train:validation:test = 80:10:10 for each downstream task dataset to evaluate the performance. For the consistency of the input graphs in pretraining and fine-tuning, we exclude implicit hydrogen atoms of molecules in fine-tuning dataset for 2D-FragCL and other reproduced 2D baselines and we include implicit hydrogen atoms of molecules in fine-tuning dataset for FragCL and 3D-InfoMax. Experimental detail follows GraphMVP (Liu et al., 2022); we fine-tune a pretrained 2D GNN with an initialized linear projection layer for 100 epochs with Adam optimizer and a learning rate of 0.001, and dropout probability of 0.5. Our results are calculated by the test ROC-AUC score of the epoch with the best validation ROC-AUC score. Besides the ROC-AUC score of individual downstream tasks, we also report the average ROC-AUC score across downstream datasets.

**Evaluation on ATOM3D downstream tasks.** Following (Townshend et al., 2020), we split the molecules in ATOM3D SMP (Small Molecule Properties) dataset into 103,547 molecules for training, 12,943 molecules for validation, and 12,943 molecules for test. We normalize the regression label by the mean and the standard deviation of the labels of training set. Our result is calculated by the test MAE score of the epoch with the best validation MAE score. We pretrain on ATOM3D training dataset and fine-tune a pretrained 2D GNN with an initialized linear projection layer for 100 epochs with Adam optimizer and Reduce-LR-On-Plateau scheduler with reduction parameter 0.7, and patience 3, and initial learning rate 0.001.

**Hardwares.** For 2D-FragCL, we use a single NVIDIA GeForce RTX 2080 Ti GPU and 40 CPU cores (Intel(R) Xeon(R) CPU E5-2630 v4 @ 2.20GHz), and a single NVIDIA-A100-SXM GPU and 124 CPU cores (AMD EPYC 7542 32-Core Processor) for FragCL. Experiments for ZINC15 dataset were performed using a single GPU (Tesla V100) and 32 CPU cores (Intel Xeon Gold 5120).

**Baselines.** We compare our method with an extensive list of baseline methods in the literature of graph representation learning:

- *No pretraining* trains a model from scratch for downstream task.
- *EdgePred* (Hamilton et al., 2017) uses edge-reconstruction as a pretext task.
- *AttrMask* (Hu et al., 2019) learns representation by recovering the vertex features after masking them.

---

[3]https://github.com/chao1224/GraphMVP
[4]https://github.com/Shen-Lab/GraphCL

Table 4: Comparison of test ROC-AUC (%) score on downstream molecular property prediction benchmarks when pretrained on ZINC15 dataset. Following the baselines, we report mean and standard deviation over 10 different seeds. We mark the best mean score bold. Additionally, we mark the second-best score bold when its mean is within one standard deviation from the highest score.

| Pretraining | BBBP | Tox21 | ToxCast | Sider | Clintox | MUV | HIV | Bace | Avg |
|---|---|---|---|---|---|---|---|---|---|
| - | $65.8_{\pm4.5}$ | $74.0_{\pm0.8}$ | $63.4_{\pm0.6}$ | $57.3_{\pm1.6}$ | $58.0_{\pm4.4}$ | $71.8_{\pm2.5}$ | $75.3_{\pm1.9}$ | $70.1_{\pm5.4}$ | 67.0 |
| Infomax | $68.8_{\pm0.8}$ | $75.3_{\pm0.5}$ | $62.7_{\pm0.4}$ | $58.4_{\pm0.8}$ | $69.9_{\pm3.0}$ | $75.3_{\pm2.5}$ | $76.0_{\pm0.7}$ | $75.9_{\pm1.6}$ | 70.3 |
| EdgePred | $67.3_{\pm2.4}$ | $76.0_{\pm0.6}$ | $\mathbf{64.1}_{\pm0.6}$ | $60.4_{\pm0.7}$ | $64.1_{\pm3.7}$ | $74.1_{\pm2.1}$ | $76.3_{\pm1.0}$ | $79.9_{\pm0.9}$ | 70.3 |
| AttrMask | $64.3_{\pm2.8}$ | $\mathbf{76.7}_{\pm0.4}$ | $\mathbf{64.2}_{\pm0.5}$ | $\mathbf{61.0}_{\pm0.7}$ | $71.8_{\pm4.1}$ | $74.7_{\pm1.4}$ | $77.2_{\pm1.1}$ | $79.3_{\pm1.6}$ | 71.2 |
| ContextPred | $68.0_{\pm2.0}$ | $75.7_{\pm0.7}$ | $63.9_{\pm0.6}$ | $60.9_{\pm0.6}$ | $65.9_{\pm3.8}$ | $\mathbf{75.8}_{\pm1.7}$ | $77.3_{\pm1.0}$ | $79.6_{\pm1.2}$ | 70.9 |
| GraphCL | $69.7_{\pm0.7}$ | $73.9_{\pm0.7}$ | $62.4_{\pm0.6}$ | $60.5_{\pm0.9}$ | $76.0_{\pm2.7}$ | $69.8_{\pm2.7}$ | $\mathbf{78.5}_{\pm1.2}$ | $75.4_{\pm1.4}$ | 70.8 |
| JOAO | $70.2_{\pm1.0}$ | $75.0_{\pm0.3}$ | $62.9_{\pm0.5}$ | $60.0_{\pm0.8}$ | $\mathbf{81.3}_{\pm2.5}$ | $71.7_{\pm1.4}$ | $76.7_{\pm1.2}$ | $77.3_{\pm0.5}$ | 71.9 |
| JOAOv2 | $\mathbf{71.4}_{\pm0.9}$ | $74.3_{\pm0.6}$ | $63.2_{\pm0.5}$ | $60.5_{\pm0.7}$ | $81.0_{\pm1.6}$ | $73.7_{\pm1.0}$ | $77.5_{\pm1.2}$ | $75.5_{\pm1.3}$ | 72.1 |
| 2D-FragCL (Ours) | $67.6_{\pm1.7}$ | $76.1_{\pm0.7}$ | $63.6_{\pm0.3}$ | $\mathbf{61.3}_{\pm0.9}$ | $80.0_{\pm2.1}$ | $\mathbf{75.4}_{\pm1.6}$ | $\mathbf{78.1}_{\pm1.3}$ | $\mathbf{81.2}_{\pm1.2}$ | **72.9** |

- *GPT-GNN* (Hu et al., 2020) uses the graph generation task as a pretext task.

- *Infomax* (Sun et al., 2019) maximizes mutual information between global representations (i.e., graph representations) and local representations (i.e. path representation).

- *ContextPred* (Hu et al., 2019) learns representation by predicting surrounding subgraph of specific node edge.

- *GraphLoG* (Xu et al., 2021) discriminates graph and subgraph pairs from their opposing pairs to preserve local similarity between various graphs, which leads to the embedding alignment of correlated graphs.

- *G-Contextual* (Rong et al., 2020a) learns representations by randomly masking local subgraphs of target nodes (or edges) and predicting these contextual properties from node embeddings.

- *G-Motif* (Rong et al., 2020a) predicts the occurrence of the semantic motifs extracted by using chemical prior.

- *GraphCL* (You et al., 2020) is a generic graph contrastive learning method based on their graph-agnostic augmentation schemes, which do not use any molecule-specific knowledge.

- *JOAO* (You et al., 2021) and *JOAOv2* (You et al., 2021) propose min-max optimization processes to learn optimal data augmentation strategies dynamically from a pre-fixed candidate set of augmentations.

- *MGSSL* (Zhang et al., 2021) introduces a generative self-supervised objective to reconstruct a motif-tree.

- *MolCLR* (Wang et al., 2021) performs a contrastive learning with NT-Xent (Chen et al., 2020a), constructing positive views of a molecule by proposed molecule augmentation schemes.

- *3D-InfoMax* (Stärk et al., 2022) proposes to consider 2D topological molecule graph and 3D geometric molecule graph from the same molecule as a positive view of each other.

# B ADDITIONAL RESULTS OF 2D-FRAGCL

To further emphasize the effectiveness of our view construction strategy, we report the performance of 2D-FragCL on the several downstream tasks on MoleculeNet (Wu et al., 2018) when pretrained on ZINC15 dataset with baselines. In Table 4, we observe that FragCL significantly improves non-pretrained counterpart. Also, FragCL achieves at least the second-best ROC-AUC score for 5 out of 8 tasks, FragCL achieves an average ROC-AUC of 72.9, while second-best baseline (i.e., JOAOv2) performs 72.1. As shown in Table 4, JOAOv2 (with respect to average ROC-AUC) often performs even worse than the vanilla without using self-supervised pretraining, e.g., in the ToxCast task. However, FragCL does not have such a failure case and achieves the robust performance across all tested downstream tasks. Overall, FragCL achieves an average ROC-AUC of 72.9, while second-best baseline (i.e., JOAOv2) performs 72.1, which indeed demonstrates the generalization ability of our FragCL.

## C  GRAPH NEURAL NETWORKS

**Graph Isomorphism Network (GIN).** We provide a detailed description of architecture of graph isomorphism network (GIN) (Xu et al., 2019), which we mainly consider as the feature extractor $f_{\text{2D}}(\cdot)$ in this paper. Particularly, GIN learns representation $\mathbf{h}_v^{(\ell)}$ by:

$$\mathbf{h}_v^{(\ell)} = \text{MLP}^{(\ell)}\big(\mathbf{h}_v^{(\ell-1)} + \sum_{u \in \mathcal{N}(v)} \big(\mathbf{h}_u^{(\ell-1)} + \mathbf{e}_{uv}^{(\ell-1)}\big)\big), \tag{15}$$

while $\mathbf{e}_{uv}^{(\ell-1)}$ is the embedding corresponding to the attribute of edge $\{u, v\} \in \mathcal{E}$.

**SchNet.** We consider SchNet (Schütt et al., 2017), which is a very strong 3D graph neural network under fair comparison (Liu et al., 2022) as our $f_{\text{3D}}(\cdot)$ in this paper.

## D  DATASET DETAILS

We pretrain our feature extractor on GEOM (Axelrod & Gomez-Bombarelli, 2022) and ZINC15 (Sterling & Irwin, 2015). Then, we perform transfer-learning on eight benchmark binary classification datasets from MoleculeNet (Wu et al., 2018). More information on downstream tasks is described below, and the statistics are reported in Table 5 and 6.

- *BBBP* contains data on whether the compound is permeable to the blood-brain barrier.
- *Tox21* measures the toxicity of a compound and was used in the 2014 Tox21 Data Challenge.
- *ToxCast* includes multiple toxicity annotations of compounds collected after performing high-throughput screening tests.
- *Sider* refers to side effect resources, i.e., data on the marketed drugs and their side effects.
- *Clintox* is a dataset of comparison results between drugs approved through the FDA and drugs removed because of toxicity during clinical trials.
- *MUV* is a validation dataset of virtual screening technology. Specifically, it is subsampled in the PubChem BioAssay using refined nearest neighborhood analysis.
- *HIV* consists of data about capability to prevent HIV replication.
- *Bace* is collected dataset of compounds that could prevent (BACE-1).

Table 5: Pretraining dataset statistics

| Dataset | GEOM | ZINC15 |
|---|---|---|
| Number of molecules | 50,000 | 2,000,000 |
| Avg. Node | 25.35 | 26.62 |
| Avg. Degree | 54.72 | 57.72 |

Table 6: MoleculeNet downstream classification dataset statistics

| Dataset | BBBP | Tox21 | ToxCast | Sider | Clintox | MUV | HIV | Bace |
|---|---|---|---|---|---|---|---|---|
| Number of molecules | 2,039 | 7,831 | 8,575 | 1,427 | 1,478 | 93,087 | 41,127 | 1,513 |
| Number of tasks | 1 | 12 | 617 | 27 | 2 | 17 | 1 | 1 |
| Avg. Node | 24.06 | 18.57 | 18.78 | 33.64 | 26.15 | 24.23 | 25.51 | 34.08 |
| Avg. Degree | 51.90 | 38.58 | 38.52 | 70.71 | 55.76 | 52.55 | 54.93 | 73.71 |

We also perform transfer-learning on seven benchmark binary classification datasets from ATOM3D (Townshend et al., 2020). More information on downstream tasks is described below, and the statistics are reported in Table 7 and 8.

Table 7: ATOM3D downstream regression tasks

| Task | Summary | Unit |
|---|---|---|
| ZPVE | Zero point vibrational energy | meV |
| $\mu$ | Dipole moment | $D$ |
| $\alpha$ | Isotropic polarizability | $bohr^3$ |
| $C_v$ | Heat capacity at $298.15K$ | $cal/mol \cdot K$ |
| LUMO | Lowest unoccupied molecular orbital energy | meV |
| HOMO | Highest occupied molecular orbital energy | meV |
| $\varepsilon_{gap}$ | Gap between HOMO and LUMO | meV |
| $R^2$ | Electronic spatial extent | $bohr^2$ |
| $U_0$ | Internal energy at $0K$ | meV |
| $U_{298}$ | Internal energy at $0K$ | meV |
| $H_{298}$ | Enthalpy at $0K$ | meV |
| $G_{298}$ | Gibbs energy at $0K$ | meV |

Table 8: ATOM3D downstream dataset statistics

| Dataset | ATOM3D |
|---|---|
| Number of molecules | 129,433 |
| Avg. Node | 18.0 |
| Avg. Degree | 18.6 |

# E  ABLATION DETAILS

Table 9: Detailed results on ablation of $\alpha$ with the average score on MoleculeNet.

| Complete bag of fragments | Incomplete bag of fragments | Disconnecting bond | Pretraining loss | BBBP | Tox21 | ToxCast | Sider | Clintox | MUV | HIV | Bace | Avg |
|---|---|---|---|---|---|---|---|---|---|---|---|---|
| - | - | - | - | $65.8_{\pm4.5}$ | $74.0_{\pm0.8}$ | $63.4_{\pm0.6}$ | $57.3_{\pm1.6}$ | $58.0_{\pm4.4}$ | $71.8_{\pm2.5}$ | $75.3_{\pm1.9}$ | $70.1_{\pm5.4}$ | 67.0 |
| Neutral | Positive | Random | $\mathcal{L}_{2D,con}$ | $65.4_{\pm1.4}$ | $73.8_{\pm0.4}$ | $63.3_{\pm0.4}$ | $57.3_{\pm1.1}$ | $74.4_{\pm4.4}$ | $74.2_{\pm1.8}$ | $75.0_{\pm0.8}$ | $79.8_{\pm1.4}$ | 70.4 |
| Positive | Positive | Random | $\mathcal{L}_{2D,con}$ | $72.2_{\pm0.7}$ | $74.6_{\pm0.7}$ | $63.1_{\pm0.3}$ | $58.6_{\pm0.8}$ | $77.9_{\pm2.3}$ | $76.5_{\pm1.7}$ | $73.6_{\pm1.4}$ | $77.5_{\pm1.3}$ | 71.7 |
| Positive | Negative | Random | $\mathcal{L}_{2D,con}$ | $71.0_{\pm0.6}$ | $73.8_{\pm0.6}$ | $63.3_{\pm0.2}$ | $59.9_{\pm0.2}$ | $81.5_{\pm3.7}$ | $75.7_{\pm2.2}$ | $75.1_{\pm1.3}$ | $76.4_{\pm3.1}$ | 72.1 |
| Positive | Negative | Single | $\mathcal{L}_{2D,con}$ | $68.2_{\pm3.0}$ | $74.7_{\pm0.3}$ | $62.4_{\pm0.7}$ | $60.3_{\pm0.6}$ | $82.7_{\pm1.3}$ | $77.9_{\pm1.0}$ | $76.1_{\pm1.4}$ | $77.2_{\pm1.0}$ | 72.4 |
| Positive | Neutral | Single | $\mathcal{L}_{con}$ | $67.9_{\pm2.4}$ | $74.5_{\pm0.1}$ | $63.8_{\pm0.4}$ | $58.2_{\pm0.7}$ | $80.5_{\pm4.4}$ | $76.7_{\pm2.3}$ | $75.4_{\pm0.2}$ | $78.1_{\pm1.3}$ | 71.9 |
| Positive | Negative | Single | $\mathcal{L}_{2D,con}$ | $68.2_{\pm3.0}$ | $74.7_{\pm0.3}$ | $62.4_{\pm0.7}$ | $60.3_{\pm0.6}$ | $82.7_{\pm1.3}$ | $77.9_{\pm1.0}$ | $76.1_{\pm1.4}$ | $77.2_{\pm1.0}$ | 72.4 |
| Positive | Negative | Random | $\mathcal{L}_{con}$ | $66.4_{\pm1.6}$ | $75.7_{\pm1.6}$ | $63.7_{\pm0.5}$ | $60.7_{\pm1.1}$ | $89.6_{\pm3.1}$ | $73.8_{\pm1.1}$ | $76.1_{\pm1.2}$ | $80.6_{\pm2.0}$ | 73.3 |
| Positive | Negative | Single | $\mathcal{L}_{con}$ | $68.4_{\pm1.4}$ | $76.3_{\pm1.2}$ | $63.5_{\pm0.4}$ | $61.2_{\pm0.6}$ | $90.5_{\pm2.4}$ | $76.3_{\pm1.0}$ | $75.0_{\pm0.7}$ | $80.8_{\pm1.0}$ | 74.0 |
| Positive | Negative | Single | $\mathcal{L}_{FragCL}$ | $70.9_{\pm1.6}$ | $76.2_{\pm0.2}$ | $64.2_{\pm0.5}$ | $61.9_{\pm0.9}$ | $89.9_{\pm1.2}$ | $77.8_{\pm0.6}$ | $77.8_{\pm0.5}$ | $80.9_{\pm1.0}$ | 75.0 |

Table 10: Detailed results on ablation of FragCL with the average score on MoleculeNet.

| $\alpha$ | BBBP | Tox21 | ToxCast | Sider | Clintox | MUV | HIV | Bace | Avg |
|---|---|---|---|---|---|---|---|---|---|
| 0.0 | $68.4_{\pm1.4}$ | $76.3_{\pm1.2}$ | $63.5_{\pm0.4}$ | $61.2_{\pm0.6}$ | $90.5_{\pm2.4}$ | $76.3_{\pm1.0}$ | $75.0_{\pm0.7}$ | $80.8_{\pm1.0}$ | 74.0 |
| 0.5 | $69.1_{\pm0.5}$ | $76.0_{\pm0.2}$ | $63.5_{\pm0.2}$ | $60.2_{\pm0.9}$ | $92.4_{\pm1.2}$ | $77.3_{\pm1.5}$ | $77.6_{\pm1.7}$ | $80.1_{\pm0.9}$ | 74.5 |
| 1.0 | $70.9_{\pm1.6}$ | $76.2_{\pm0.2}$ | $64.2_{\pm0.5}$ | $61.9_{\pm0.9}$ | $89.9_{\pm1.2}$ | $77.8_{\pm0.6}$ | $77.8_{\pm0.5}$ | $80.9_{\pm1.0}$ | 75.0 |
| 2.0 | $69.2_{\pm0.2}$ | $75.3_{\pm0.7}$ | $64.0_{\pm0.3}$ | $60.3_{\pm0.1}$ | $92.0_{\pm2.7}$ | $76.6_{\pm1.4}$ | $77.2_{\pm0.6}$ | $79.6_{\pm0.5}$ | 74.3 |

## F   DISCUSSION ABOUT THE FRAGMENTATION STRATEGY

The extent of semantic-preservation can be controlled by a more sophisticated fragmentation strategy to control. In Table 11, we report the performance with the fragmentation strategy 'disconnect a non-ring C-C bond' to prevent altering the named groups such as ether, ester, and amide. In the table below, modified strategy obtains improvements on some downstream dataset (e.g., $89.9 \rightarrow 92.1$ in Clintox). However, we note that our original (and extremely simple) strategy also shows comparable performance for downstream tasks, verifying the validity of 'disconnect a non-ring single bond' as a semantic-preserving transformation.

Table 11: Detailed results on the fragmentation strategy of FragCL with the average score on MoleculeNet.

| Fragmentation strategy | BBBP | Tox21 | ToxCast | Sider | Clintox | MUV | HIV | Bace | Avg |
|---|---|---|---|---|---|---|---|---|---|
| Single bond | $70.9_{\pm 1.6}$ | $76.2_{\pm 0.2}$ | $64.2_{\pm 0.5}$ | $61.9_{\pm 0.9}$ | $89.9_{\pm 1.2}$ | $77.8_{\pm 0.6}$ | $77.8_{\pm 0.5}$ | $80.9_{\pm 1.0}$ | 75.0 |
| C-C single bond | $70.4_{\pm 0.7}$ | $75.6_{\pm 0.8}$ | $64.6_{\pm 0.6}$ | $61.5_{\pm 1.2}$ | $92.1_{\pm 1.9}$ | $77.5_{\pm 1.6}$ | $77.6_{\pm 1.2}$ | $80.4_{\pm 1.7}$ | 75.0 |

## G   RESULTS ON SPHERENET AS A 3D ENCODER

Our architectural choice of Table 1 is for the fair comparison with GraphMVP (one of our main baselines). In Table 12, we verify our torsional angle reconstruction task is still meaningful even with a torsion-angle-aware 3D encoder, i.e., SphereNet (Liu et al., 2021), by improving the performance by $74.4 \rightarrow 75.1$.

Table 12: Detailed results on SphereNet as a 3D encoder of FragCL with the average score on MoleculeNet.

| Pretraining loss | BBBP | Tox21 | ToxCast | Sider | Clintox | MUV | HIV | Bace | Avg |
|---|---|---|---|---|---|---|---|---|---|
| $\mathcal{L}_{con}$ | $69.8_{\pm 1.3}$ | $75.6_{\pm 0.3}$ | $64.9_{\pm 0.6}$ | $62.0_{\pm 1.1}$ | $89.9_{\pm 1.7}$ | $75.9_{\pm 0.4}$ | $76.4_{\pm 0.3}$ | $80.3_{\pm 0.9}$ | 74.4 |
| $\mathcal{L}_{FragCL}$ | $71.5_{\pm 1.4}$ | $75.7_{\pm 0.7}$ | $65.7_{\pm 0.3}$ | $61.3_{\pm 1.0}$ | $91.4_{\pm 1.4}$ | $78.1_{\pm 2.2}$ | $76.1_{\pm 1.6}$ | $80.7_{\pm 0.6}$ | 75.1 |

# H    TRAINING PROCEDURE OF FRAGCL

---

**Algorithm 1** Fragment-based molecule Contrastive Learning (FragCL)

---

**Input:**  Molecule graphs $\{\mathcal{G}\}_{i=1}^{M}$, a 2D GNN $f_{\text{2D}}(\cdot)$, 3D GNN $f_{\text{3D}}(\cdot)$, and projection heads $g_{\text{2D}}(\cdot)$, $g_{\text{3D}}(\cdot)$, $g_{\text{rot}}(\cdot)$, and $g_{\text{abs}}(\cdot)$.

---

1: **for** sampled mini-batch of $N$ molecule graphs $\{\mathcal{G}_i\}_{i=1}^{N}$ **do**
2:     **for** $i = 1$ **to** $N$ **do**
3:         $\{\mathcal{G}'_{\text{2D},i}, \mathcal{G}''_{\text{2D},i}\}, \{\mathcal{G}'_{\text{3D},i}, \mathcal{G}''_{\text{3D},i}\} \leftarrow \texttt{Fragmentation}(\mathcal{G}_i)$
4:         // Obtain 2D graph representations
5:         $\mathbf{r}_{\text{2D},i}, \mathbf{r}'_{\text{2D},i}, \mathbf{r}''_{\text{2D},i}, \leftarrow f_{\text{2D}}(\mathcal{G}_{\text{2D}}), f_{\text{2D}}(\mathcal{G}'), f_{\text{2D}}(\mathcal{G}''),$
6:         $\mathbf{r}_{\text{2D},i}^{\texttt{mix}} \leftarrow \frac{1}{n'_i + n''_i}(n'_i \times f_{\text{2D}}(\mathcal{G}') + n''_i \times f_{\text{2D}}(\mathcal{G}''))$
7:         $\mathbf{z}_{\text{2D},i}, \mathbf{z}'_{\text{2D},i}, \mathbf{z}''_{\text{2D},i}, \mathbf{z}_{\text{2D},i}^{\texttt{mix}} \leftarrow g_{\text{2D}}(\mathbf{r}_{\text{2D},i}), g_{\text{2D}}(\mathbf{r}'_{\text{2D},i}), g_{\text{2D}}(\mathbf{r}''_{\text{2D},i}), g_{\text{2D}}(\mathbf{r}_{\text{2D},i}^{\texttt{mix}})$
8:         // Obtain 3D graph representations
9:         $\mathbf{r}_{\text{3D},i}, \mathbf{r}'_{\text{3D},i}, \mathbf{r}''_{\text{3D},i} \leftarrow f_{\text{3D}}(\mathcal{G}_{\text{3D}}), f_{\text{3D}}(\mathcal{G}'), f_{\text{3D}}(\mathcal{G}'')$
10:        $\mathbf{r}_{\text{3D},i}^{\texttt{mix}} \leftarrow \frac{1}{n'_i + n''_i}(n'_i \times f_{\text{3D}}(\mathcal{G}') + n''_i \times f_{\text{3D}}(\mathcal{G}''))$
11:        $\mathbf{z}_{\text{3D},i}, \mathbf{z}'_{\text{3D},i}, \mathbf{z}''_{\text{3D},i}, \mathbf{z}_{\text{3D},i}^{\texttt{mix}} \leftarrow g_{\text{3D}}(\mathbf{r}_{\text{3D},i}), g_{\text{3D}}(\mathbf{r}'_{\text{3D},i}), g_{\text{3D}}(\mathbf{r}''_{\text{3D},i}), g_{\text{3D}}(\mathbf{r}_{\text{3D},i}^{\texttt{mix}})$
12:        // Contrastive objectives
13:        **define** $\mathcal{L}_{\text{2D,con},i} := \mathcal{L}_{\text{NT-Xent}}(\mathbf{z}_{\text{2D},i}, \mathbf{z}_{\text{2D},i}^{\texttt{mix}}, \{\mathbf{z}_{\text{2D},j}^{\texttt{mix}}\}_{j \neq i} \cup \{\mathbf{z}'_{\text{2D},i}, \mathbf{z}''_{\text{2D},i}\})$
14:        **define** $\mathcal{L}_{\text{3D,con},i} := \mathcal{L}_{\text{NT-Xent}}(\mathbf{z}_{\text{3D},i}, \mathbf{z}_{\text{3D},i}^{\texttt{mix}}, \{\mathbf{z}_{\text{3D},j}^{\texttt{mix}}\}_{j \neq i} \cup \{\mathbf{z}'_{\text{3D},i}, \mathbf{z}''_{\text{3D},i}\})$
15:        **define** $\mathcal{L}_{\{\text{2D,3D}\},\text{con},i} := \frac{1}{2}\big(\mathcal{L}_{\text{NT-Xent}}(\mathbf{z}_{\text{2D},i}, \mathbf{z}_{\text{3D},i}, \{\mathbf{z}_{\text{3D},j}\}_{j=1}^{n}) + \mathcal{L}_{\text{NT-Xent}}(\mathbf{z}_{\text{3D},i}, \mathbf{z}_{\text{2D},i}, \{\mathbf{z}_{\text{2D},j}\}_{j=1}^{n})\big)$
16:        **define** $\mathcal{L}_{\text{con},i} := \mathcal{L}_{\text{2D,con},i} + \mathcal{L}_{\text{3D,con},i} + \mathcal{L}_{\{\text{2D,3D}\},\text{con},i}$
17:        // Torsional angle reconstruction of $(s, u, v, t)$ around the fragmented edge $(u, v)$
18:        $\mathbf{z}_{\text{rot},i}(s, u, v, t) \leftarrow g_{\text{rot}}([\mathbf{h}_{\text{2D},s,i}; \mathbf{h}_{\text{2D},u,i}; \mathbf{h}_{\text{2D},v,i}; \mathbf{h}_{\text{2D},t,i}])$
19:        $\mathbf{z}_{\text{abs},i}(s, u, v, t) \leftarrow g_{\text{abs}}([\mathbf{h}_{\text{2D},s,i}; \mathbf{h}_{\text{2D},u,i}; \mathbf{h}_{\text{2D},v,i}; \mathbf{h}_{\text{2D},t,i}])$
20:        $\mathbf{z}_{\text{abs},i}(t, v, u, s) \leftarrow g_{\text{abs}}([\mathbf{h}_{\text{2D},t,i}; \mathbf{h}_{\text{2D},v,i}; \mathbf{h}_{\text{2D},u,i}; \mathbf{h}_{\text{2D},s,i}])$
21:        $\mathbf{z}_{\text{abs},i}(t, v, u, s) \leftarrow g_{\text{abs}}([\mathbf{h}_{\text{2D},t,i}; \mathbf{h}_{\text{2D},v,i}; \mathbf{h}_{\text{2D},u,i}; \mathbf{h}_{\text{2D},s,i}])$
22:        $\mathcal{L}_{\text{rot},i} \leftarrow \mathcal{L}_{\text{BCE}}(\mathbf{z}_{\text{rot},i}(s, u, v, t), y_{\text{rot},i}) + \mathcal{L}_{\text{BCE}}(\mathbf{z}_{\text{rot},i}(t, v, u, s), y_{\text{rot},i})$
23:        $\mathcal{L}_{\text{abs},i} \leftarrow \mathcal{L}_{\text{BCE}}(\mathbf{z}_{\text{abs},i}(s, u, v, t), y_{\text{rot},i}) + \mathcal{L}_{\text{BCE}}(\mathbf{z}_{\text{abs},i}(t, v, u, s), y_{\text{rot},i})$
24:        **define** $\mathcal{L}_{\text{torsion},i} := \mathcal{L}_{\text{rot},i} + \mathcal{L}_{\text{abs},i}$
25:        // Definition of FragCL loss
26:        **define** $\mathcal{L}_{\text{FragCL},i} := \mathcal{L}_{\text{con},i} + \alpha \mathcal{L}_{\text{torsion},i}$
27:     **endfor**
28:     **define** $\mathcal{L}_{\text{FragCL}} := \frac{1}{n}\sum_{i=1}^{n}\mathcal{L}_{\text{FragCL},i}$
29:     Update $f_{\theta}(\cdot)$ and $g_{\phi}(\cdot)$ to minimize $\mathcal{L}_{\text{FragCL}}$
30: **endfor**
31: **return** $f_{\theta}(\cdot)$

---

## I FRAGMENTATION EXAMPLES

We present several examples of fragmentation strategy based on our algorithm. Our fragmentation scheme almost preserves the chemically informative substructures, keeping the semantics of the original molecule.

Table 13: Fragmentation examples

| Molecule | Bag of Fragments |
|----------|------------------|

