# OpenReview forum: "Contrastive Learning of Molecular Representation with Fragmented Views"
_ICLR.cc/2023/Conference — Submitted to ICLR 2023_

### Official Review · Reviewer_EnEZ · 2022-10-20

**Confidence:** 2
**Correctness:** 4
**Technical Novelty And Significance:** 2
**Empirical Novelty And Significance:** 2
**Recommendation:** 5

**Clarity, Quality, Novelty And Reproducibility:**

The paper is easy to follow and clearly written. The proposed approach to constructing the positive and negative view of molecules and the idea of incorporating  3D structure by reconstructing torsional angle seem to be novel.
With the details in the appendix, the paper seems to be reproducible.  However, I feel like there is a bit of repetition in the main text about the proposed method around positive and negative views while many modeling details are left in the appendix.

**Strength And Weaknesses:**

Strength:
The paper addresses an interesting problem such as how to define a good positive and negative view when we apply contrastive learning on the molecule. They also propose an interesting idea that seems to be an easy direct way to incorporate 3D information about molecules.

Weakness:
Experimental results seem weak especially when trained on 2D molecule graphs.
Compare to the existing baselines, the proposed scheme to construct a negative and positive view of molecules does not seem to have many advantages.

**Summary Of The Paper:**


The paper proposes a fragment-based contrastive learning algorithm for learning the discriminative representations of the molecules. They propose to decompose a molecule into two fragments by breaking a single bound (none ring). The bound is chosen such that two fragments have a similar number of atoms. Then the complete bag of fragments of the molecule is considered as a positive view while the incomplete or the complete bag of fragments of the other molecule is considered as a negative view. They also  Incorporated 3D structure by including a reconstruction loss on the torsional angle.

**Summary Of The Review:**

Overall, the paper is clearly written and easy to follow, however, I have some detailed questions as following:

1. How often does the assumption " breaking a none-ring single bound preserve the  chemically informative substructures"

2, Is the negative view which is the complete fragments of another molecule really useful, seems to be a very obvious negative view that is easy to discriminate.

3. It is not very clear here: "Note that the semantic of a fragment g_i is ensured to be sufficiently different from the original molecule g'_i  since ....... ".  According to the paper, F(g'_j) and F(g''_j)  are complementary, not necessarily similar. And I would think also that the negative view which is constructed by the incomplete bag of fragments is not hard to discriminate from science dropping half of the molecule could change the functional group of the original molecule drastically.

4. When looking at the experimental results in Table1, it feels to me that the model did not have much advantage over the other models when working on 2D representation,  but it shows better performance when trained on both 2D and 3D. Could it be possible that the improvement comes from the fact that the model also tries to incorporate the reconstruction of the torsional angle but less due to introduced strategies about constructing a positive and negative view of the molecule?

5. The ablation study section is a bit confusing to me, so the author first used the incomplete fragment as a positive view, and on top of that used the complete fragments as a positive view, and then on top of that used the incomplete fragment as a negative view? Also, it would be interesting to see what is the effect of the complete fragments of another molecule as a negative view.

6. As table 3 shows, when all settings are kept the same and only changing where we cut seems not to have such a big difference in terms of performance,e 72.1 to 72.4 (on average), instead adding the loss on torsion angle seem to improve a lot (74 to 75).

7. It would be great to see what the value of alpha is in the best model, and what happens if we set alpha to zero which means there is no reconstruction of the torsional angle

---

> ### Author Response · Authors · 2022-11-14
> **Response to Reviewer EnEZ (2/2)**
>
> **(Q5) The improvement comes from the reconstruction of the torsional angle but less due to strategies about constructing positive and negative views?**
>
> The improvement comes from both the torsion reconstruction task and the construction strategies of positive and negative views. The table below shows that our view construction strategy (denoted by $\mathcal{L}_{\mathtt{con}}$, Eq(9)) already outperforms prior arts 3D-Infomax by 73.4 -> 74.0 on MoleculeNet, verifying the effectiveness of the view construction strategy. We added this respect of discussion in Section 4.3.
>
> \begin{array}{l|ccc}
> \hline
> & \text{3D-Infomax} & \mathcal{L}_\mathtt{con} & \mathcal{L}_\mathtt{FragCL} \newline
> \hline
> \text{Avg.} & 73.4 & 74.0 & 75.0 \newline
> \hline
> \end{array}
>
> ---------
> **(Q6) Clarification of ablation study.**
>
> Each row of Table 3 is independent. All settings are as indicated in the specific row. For the effect of the complete fragments of another molecule as a negative view, please see Q2.
>
> ----------
> **(Q7) Cutting strategy seems not to have much improvement.**
>
> The cutting strategy is effective enough since it positively affects not only 2D molecular representations, but also 3D ones. To support this, we conduct an additional ablation experiment. As shown in the table below, when removing our strategy in both 2D and 3D contrastive learning, we lose 0.7 average performance (74.0->73.3) on MoleculeNet, which highlights the strength of our strategy. Note that the gain from our cutting strategy is similar to that from torsional angle prediction (74.0->75.0).
>
> \begin{array}{cccc|c}
> \hline
> \text{Complete bag of fragments} & \text{Incomplete bag of fragments} & \text{Disconnecting bond} & \text{Pretraining loss} & \text{Avg.}\newline
> \hline
> \text{Positive} & \text{Negative} & \text{Random} & \mathcal{L}_\mathtt{con} & 73.3 \newline
> \text{Positive} & \text{Negative} & \text{Single} & \mathcal{L}_\mathtt{con} & 74.0 \newline
> \text{Positive} & \text{Negative} & \text{Single} & \mathcal{L}_\mathtt{FragCL} & 75.0 \newline
> \hline
> \end{array}
>
> --------
> **(Q8) Choice of $\alpha$.**
>
> We found that the optimal $\alpha$ is 1 in our experiment. Below is the ablation study on $\alpha$ on MoleculeNet. Also, we note that the result of $\alpha=0$ is already reported in Table 3, which coincides with pretraining loss $\mathcal{L}_{\mathtt{con}}$. We added the results of the ablation of $\alpha$ in Appendix E.
>
> \begin{array}{l|cccc}
> \hline
> \alpha & 0 & 0.5 & 1 & 2\newline
> \hline
> \text{Avg.} & 74.0 & 74.5 & 75.0 & 74.3 \newline
> \hline
> \end{array}
>
> ------
> **(Q9) Comments for the organization of manuscript.**
>
> Many thanks for the careful reading to improve the clarity of our manuscript. Following your suggestions, we carefully revised our manuscript.

---

> ### Author Response · Authors · 2022-11-14
> **Response to Reviewer EnEZ (1/2)**
>
> Dear reviewer EnEZ,
>
> We sincerely appreciate your efforts in reviewing our manuscript. We respond to each comment in the following content. We carefully incorporated the discussions into the final manuscript.
>
> ------
> **(Q1) How often does the assumption “breaking a non-ring single bond preserve the chemically informative substructures hold?**
>
> Our disconnecting strategy almost preserves the “chemically informative substructures”. The main characteristics of chemical groups are determined by higher order bonds or electronegative atoms (not by the presence of single bonds). For example, named groups such as esters, amides, and ketones are categorized by the “carbonyl group”, which consists of a C=O double bond. The main functionality of esters, amides, and ketones are mainly determined by C=O double bond and our fragmentation scheme preserves the carbonyl group (C=O double bond). Also, the chemical functionality of the named group ether and thioether comes from the electronegative oxygen and sulfur atoms (not from the single bond), respectively [1]. This verifies the validity of our assumption. We clarified this in our revision (Section 3.1).
>
> [1] Smith, “Organic Chemistry”, McGraw Hill, 6th Ed
>
> --------
> **(Q2) Is the negative view which is the complete fragments of another molecule really useful?**
>
> It is beneficial to use complete bags of fragments of other molecules as negative views. In the table below, we verify that our framework also achieves reasonable performance by 67.2 -> 71.9 on MoleculeNet compared to its non-pretraining counterpart even when we consider only “complete fragments of other molecules” as negative samples. We added this respect of discussion in Section 4.3.
>
> \begin{array}{cc|c}
> \hline
> \text{Negative samples} \newline
> \hline
> \text{Complete bag of fragments} & \text{Incomplete bag of fragments}\newline
> \text{of other molecules} & \text{of its own} & \text{Avg.}\newline
> \hline
> \text{-} & \text{-} & 67.2 \newline
> \text{O} & \text{X} & 71.9 \newline
> \hline
> \end{array}
>
>
> ------------
> **(Q3) “Note that the semantic of ….”: $F(\mathcal{G}’_j)$ and $F(\mathcal{G}’’_j)$ are similar?**
>
> Your comment seems to be from a slight misunderstanding of our paper. $F(\mathcal{G}’_j)$ and $F(\mathcal{G}’’_j)$ are not necessarily similar. The equation $|F(\mathcal{G}’_j)| \approx |F(\mathcal{G}’’_j)|$ means that the “number” of chemically informative substructures of $\mathcal{G}’_j$ and $\mathcal{G}’’_j$ are likely to be similar. Thus, $F(\mathcal{G}’_j)$ and $F(\mathcal{G}’’_j)$ need not be similar.
>
> -------
> **(Q4) The negative view which is constructed by the incomplete bag of fragments is not hard to discriminate?**
>
> The incomplete bag of fragments are harder to be discriminated against compared to the complete bag of fragments from other molecules since they have common substructures with the molecule being compared. For example, the cosine similarity of features between the original molecule and incomplete bag of fragments shows average #2.4 rank among the negative samples (complete bag of fragments of other molecules) in a mini-batch of size 256. This verifies that the incomplete bags of fragments are indeed hard-to-discriminate.

---

> ### Author Response · Authors · 2022-12-07
> **A Gentle Reminder**
>
> Dear Reviewer EnEZ,
>
> Thank you very much again for your time and efforts in reviewing our paper.
>
> We kindly remind that we have only a week or so in the discussion period.
>
> We just wonder whether there is any further concern and hope to have a chance to respond before the discussion phase ends.
>
> Many thanks, Authors

---

### Official Review · Reviewer_ov3H · 2022-10-21

**Confidence:** 5
**Correctness:** 2
**Technical Novelty And Significance:** 2
**Empirical Novelty And Significance:** 2
**Recommendation:** 3

**Clarity, Quality, Novelty And Reproducibility:**

There are some issues with the writing. In addition to some typos, the flow of the paper needs some work. For example, the torsional loss is introduced very early in the paper, but is not explained until the end of Section 3.3. Additionally, several details of the implementation are missing from the supplement. A non-comprehensive list of clarity issues are listed above.

There are also some claims and justifications for the fragmentation scheme, which are chemically dubious or incorrect (listed above).

The novelty of the work is also limited. Other works before have used many of the pieces in this paper (e.g. molecule fragmentation for contrastive learning, combining 2D and 3D GNNs, etc.). The main novelty of the paper is a different way of applying fragments to the contrastive loss.

**Strength And Weaknesses:**

The paper proposes a novel scheme for fragmentation (as opposed to previous fragmentation schemes) for contrastive learning on molecules. The scheme is interesting, and worth publicizing (at least on arXiv). The improvements on benchmarks are also nice to see.

Below are some weaknesses which I believe need to be addressed before publication:
### The complexity of the objective suggests a lack of robustness
In addition to the fact that there are two separate GNNs, there are a lot of loss functions in the objective (i.e. contrastive loss for the 2D GNN, contrastive loss for the 3D GNN, contrastive losses between the two GNNs, and a complex loss function for the torsion angle which consists of both regression and multiclass classification objectives. Not only does this greatly increase training and inference time (and computational resource use), it also casts doubt on the robustness of the neural network. Especially because the improvements in downstream prediction tasks are only modest (if present at all), it suggests that all this extra complexity may be the result of overfitting to these specific benchmarks.

For example, it is difficult to justify predicting the torsion angle both as a regression problem and as a multiclass classification problem (over bins of the angle). What happens if the binned objective is removed?

The ablation analysis also suggests that each piece of this complex framework is necessary, even for the few modest improvements.
### Incorrect chemical justification for the proposed fragmentation scheme
The authors attempt to justify the fragmentation scheme using several incorrect statements. For example, the claim is made that fragmenting a molecule along a single bond is “likely to preserve chemically informative substructures”. They also claim that “single bonds are not directly involved in reaction pathways in general”, and that the "fragmentation scheme described in Section 3.1, satisfies $F(G\_{i}) = F(G'\_{i})\cup F(G''_{i})$ with a high probability” (where $F(G)$ is the set of functional groups in $G$). These statements are all dubious if not incorrect.

Single bonds are oftentimes highly important in reaction pathways. Nucleophilic attacks such as SN1 and SN2 rely on the breaking of single bonds almost ubiquitously, not double or triple bonds as the authors claim. If anything, single bonds are perhaps even more likely to participate in reaction chemistry (compared to double or triple bonds) because the bond energy is usually lower.

Breaking single bonds also can very easily break functional groups. For example, consider dibenzyl ether. Based on the fragmentation scheme proposed by the authors, this molecule would be split into two fragments which completely destroy the ether functional group. Groups such as esters, amides, and ketones would also easily be split up. The statement that $F(G\_{i}) = F(G'\_{i})\cup F(G''_{i})$ with high probability should be quantified over the dataset if it is true (although I have very strong suspicions that it is not).

Furthermore, the distinction between single and multiple bonds in organic molecules is not as clear as the authors suggest. For example, consider a highly conjugated alkene capped with aldehydes (or another resonant functional group). Although the graph representation of this molecule will have several single bonds, the distinction between the single and double bonds in reality are actually minimal, as the entire structure is in resonance (which would be broken by fragmentation). Thus, the choice to fragment molecules by single bonds only is not particularly well justified.

It is also good to note that there are many molecules which would not satisfy the fragmentation scheme suggested by this paper, such as benzene, naphthalene, anthracene, etc. These molecules could not be fragmented at all using this scheme.
### Areas lacking in explanation, clarity, or typos
- In the penultimate sentence of Section 3.1, $G''_{2D}$ is a typo, I think
- Eq. 10 – 11 should specify what the $h$s mean. I assume they are the node representations from the last layer of the 2D GNNs
- In Table 1, the Avg column is not informative and should be removed. These are very different benchmarks with varying difficulties, and averaging over the scores doesn’t make sense. After all, a model could get a very high Avg just by optimizing very specifically for a single very easy task.
- The bolding in Table 1 and 2 is not clear. Why are there multiple numbers bolded in a single column?
- In Section 4.3, clarify what the loss value is. Is this just the NT-Xent value? If so, this metric does not make much sense because the fragmentation scheme is different between different experiments, and so the NT-Xent values will be based on different inputs and therefore are not comparable across ablation experiments
- In Appendix A, the definition of $r^{mix}_{i}$ contradicts the main text
- There are typos in the title of Appendix B, and in the Table 9 caption
- What is the form of the projectors $g$ which map from node embeddings to torsional angle? I assume they are single linear projection heads (i.e. a single linear layer) on the concatenation of node embeddings
- Once pretraining is done, how are the final tasks (e.g. the ATOM3D regression tasks) trained? Is it another linear projection head on the average node representations? This doesn’t seem to be explained anywhere
- The computation of the score of these benchmarks should be explained: how are these scores computed?

**Summary Of The Paper:**

The paper develops a method for constructing representations of small molecules using contrastive learning. For a particular molecule, the molecule is split into two equal-sized fragments by breaking a single non-ring bond. The method attempts to maximize similarity between the original molecule’s representation and the weighted average of the representations of the two fragments. It also attempts to minimize similarity between the original molecule’s representation and the individual fragments on their own or the average of other molecules’ fragments. The authors also train two separate neural networks—a 2D GNN and a 3D GNN—and encourage representations learned between the two GNNs to be similar, and also add on a loss to predict torsional angle of the cut bond.

The authors use this method to pretrain and then attempt to predict several benchmark molecular properties, comparing to other similar methods of generating molecular representations. They demonstrate modest improvements in some of the tasks.

**Summary Of The Review:**

The work is interesting, and there are some modest improvements in some of the benchmarks shown. However, there is limited novelty in the technical contributions. Additionally, the method is very complex, combining 2 GNNs and 3 – 4 distinct types of loss functions. Even with the complex architectures and training, the improvements in the benchmarks are not particularly impressive. Combined with the numerous issues with clarity and incorrect justifications of the fragmentation scheme (one of the main focuses of the paper), it is very difficult to support its passage.

---

> ### Author Response · Authors · 2022-11-14
> **Response to Reviewer ov3H (3/3)**
>
> **(Q9) Editorial comments.**
>
> Many thanks for the careful reading, and making constructive suggestions to improve the clarity of our manuscript. Following your suggestions, we carefully revised our manuscript.
>
> -----------
> **(Q10) $\mathcal{G}_{\mathtt{2D}}’’$ in the penultimate sentence of Section 3.1 is a typo?**
>
> It is not a typo. The sets of nodes in 3D fragments $\mathcal{G_{\mathtt{3D}}}'$ and $\mathcal{G_{\mathtt{3D}}}''$ are obtained by the nodes of 2D fragments $\mathcal{G_{\mathtt{2D}}}'$ and $\mathcal{G_{\mathtt{2D}}}''$, respectively.
>
> -------
> **(Q11) Avg column on Table 1.**
>
> We reported the Avg performance following GraphMVP [1]. Nevertheless, following your suggestion, we removed the Avg column in the revised draft.
>
> [1] Liu et al., “Pre-training Molecular Graph Representation with 3D Geometry”, ICLR 2022
>
> ---------
> **(Q12) Bolding in Table 1 and 2.**
>
> We mark the best mean score bold. Additionally, we mark the average scores within one standard deviation from the highest average score to be bold. To alleviate your concern, we described the bolding strategy in the revised manuscript.
>
> -------
> **(Q13) Clarify the loss value in Section 4.3.**
>
> We do not report loss value in Section 4.3. The only metric used in Section 4.3 is the average ROC-AUC score on MoleculeNet dataset (e.g., the right-most “Avg” column in Table 3).
>
> -------
> **(Q14) The form of the projectors $g$ which map from embeddings to torsional angle?**
>
> For $g_{\mathtt{rot}}$ and $g_{\mathtt{abs}}$ in equation (10) and (11), we use 2-layer and 3-layer MLPs , respectively. The first layer is shared between $g_{\mathtt{rot}}$ and $g_{\mathtt{abs}}$. We clarified  this point in Appendix A.
>
> -------
> **(Q15) How are the final downstream tasks trained?**
>
> As we mentioned in Appendix A of the original draft, we use another initialized linear projection layer on the average node representations.
>
> --------
> **(Q16) The computation score of the benchmarks.**
>
> We use ROC-AUC score on MoleculeNet downstream dataset and MAE score on ATOM3D downstream dataset. All values are based on the test scores with the highest validation scores.
>
> ---------
> **(Q17) Novelty is limited.**
>
> We politely disagree with the reviewer’s opinion. Our main novelty is on how to construct positive and negative views, which plays an essential role in contrastive learning [1]. Specifically, we consider an incomplete bag of fragments (e.g., a subgraph of molecular graph) as a negative view, which is strikingly different from prior approaches using a subgraph as a positive view [2,3]. Also, we introduce the torsional angle reconstruction task, which can incorporate 3D information to 2D molecule encoder. Overall, we do believe that our work provides novel ideas on molecule representation learning, as highlighted by Review yCZe and EnEZ.
>
> [1] You et al., “Graph Contrastive Learning Automated”, ICML 2021
>
> [2] You et al., “Graph Contrastive Learning with Augmentations”, NeurIPS 2020
>
> [3] Wang et al., “Molecular contrastive learning of representations via graph neural networks”, NMI 2022

---

> ### Author Response · Authors · 2022-11-14
> **Response to Reviewer ov3H (2/3)**
>
> **(Q5) Single bonds are perhaps even more likely to participate in reaction chemistry because the bond energy is usually lower.**
>
> This statement is not true. The bond energy is not necessarily correlated to the reactivity of a chemical bond. The bond energy is the energy needed for homolytic cleavage of a bond. However, general bond-dissociation reactions (e.g., SN1 and SN2 reactions) are triggered by the difference of the electron density, yielding heterolytic cleavage of a bond.
>
> -----------
> **(Q6) Breaking single bonds also can easily break functional groups.**
>
> Thank you for the constructive feedback. Although we agree that a single bond can be included in some functional groups (e.g., ether, ketone, ester, and amide), our fragmentation scheme still preserves the main characteristic of the mentioned functional groups. For example, we note that esters, amides, and ketones are commonly categorized by the “carbonyl group”, which consists of a C=O double bond. The main functionality of esters, amides, and ketones are mainly determined by C=O double bond and our fragmentation scheme preserves the carbonyl group (C=O double bond). Also, the chemical functionality of ether comes from the electronegative oxygen atom (not from the single bond) [1]. We revised the manuscript to correct the misleading term “functional group”.
>
> Nevertheless, one may try a more sophisticated fragmentation strategy to control the extent of semantic-preservation. For example, we tested ‘disconnect a non-ring C-C bond’ to prevent altering ether, ester, and amide. In the table below, modified strategy obtains improvements on some downstream dataset (e.g., 89.9 -> 92.1 in Clintox). However, we note that our original (and extremely simple) strategy also shows comparable performance for downstream tasks, verifying the validity of ‘disconnect a non-ring single bond’ as a semantic-preserving transformation. We added this respect of discussion in Appendix F.
>
> \begin{array}{l|cccccccc}
> \hline
> & \text{BBBP} & \text{Tox21} & \text{ToxCast} & \text{Sider} & \text{Clintox} & \text{MUV} & \text{HIV} & \text{Bace}\newline
> \hline
> \text{Single} & 70.9\small{\pm{1.6}} & 76.2\small{\pm{0.2}} & 64.2\small{\pm{0.5}} & 61.9\small{\pm{0.9}} & 89.9\small{\pm{1.2}} & 77.8\small{\pm{0.6}} & 77.8\small{\pm{0.5}} & 80.9\small{\pm{1.0}} \newline
> \text{C-C Single} & 70.4\small{\pm{0.7}} & 75.6\small{\pm{0.8}} & 64.6\small{\pm{0.6}} & 61.5\small{\pm{1.2}} & 92.1\small{\pm{1.9}} & 77.5\small{\pm{1.6}} & 77.6\small{\pm{1.2}} & 80.4\small{\pm{1.7}} \newline
> \hline
> \end{array}
>
> [1] Smith, “Organic Chemistry”, McGraw Hill, 6th Ed
>
> ---------
> **(Q7) The choice to fragment molecules by single bonds only is not well justified in resonance structures.**
>
> For resonance structures (e.g., conjugated aldehydes), we should cut only the single bond in the graph representation (not a higher order bond) for preserving resonance in fragments. If we cut a higher order bond, the resonance structures of fragments are destroyed.
>
> ----------
> **(Q8) Molecules which would not satisfy the fragmentation scheme.**
>
> We simply exclude such molecules before pretraining since there exist a very small number of the molecules: for example, only 51 molecules (0.1%) does not satisfy our fragmentation scheme in the GEOM dataset. We clarified this point in Appendix A.

---

> ### Author Response · Authors · 2022-11-14
> **Response to Reviewer ov3H (1/3)**
>
> Dear reviewer ov3H,
>
> We sincerely appreciate your efforts in reviewing our manuscript. We respond to each comment in the following content. We carefully incorporated the discussions into the final manuscript.
>
> -----------
> **(Q1) The complexity of the objective suggests a lack of robustness… it suggests that all this extra complexity may be the result of overfitting to these specific benchmarks.**
>
> We do not tune the hyperparameters to specific benchmarks and hence our result is not from overfitting to specific benchmarks. As shown in Table 1 and 2 of the original draft, our method outperforms prior works over a wide range of tasks using the same set of hyperparameters. We emphasize that we do not put much effort into tuning the results and our framework is robust to change of hyperparameters. For example, we report the results of varying the hyperparameter $\alpha$ in Eq. (14). Our method consistently outperforms the prior arts with sub-optimal choice of the hyperparameters, verifying the robustness of our objective. We added the results of the ablation of $\alpha$ in Appendix E.
>
> \begin{array}{l|cccccccc|c}
> \hline
> & \text{BBBP} & \text{Tox21} & \text{ToxCast} & \text{Sider} & \text{Clintox} & \text{MUV} & \text{HIV} & \text{Bace} & \text{Avg.}\newline
> \hline
> \text{3D-Infomax} & 67.9\small{\pm{1.2}} & 75.3\small{\pm{0.3}} & 64.6\small{\pm{0.4}} & 59.6\small{\pm{0.7}} & 89.7\small{\pm{0.5}} & 76.7\small{\pm{0.6}} & 73.4\small{\pm{1.2}} & 79.9\small{\pm{0.9}} & 73.4 \newline
> \hline
> \text{FragCL}(\alpha=0.0) & 68.4\small{\pm{1.4}} & 76.3\small{\pm{1.2}} & 63.5\small{\pm{0.4}} & 61.2\small{\pm{0.6}} & 90.5\small{\pm{2.4}} & 76.3\small{\pm{1.0}} & 75.0\small{\pm{0.7}} & 80.8\small{\pm{1.0}} &74.0 \newline
> \text{FragCL}(\alpha=0.5) & 69.1\small{\pm{0.5}}&76.0\small{\pm{0.2}}&63.5\small{\pm{0.2}}&60.2\small{\pm{0.9}}&92.4\small{\pm{1.2}}&77.3\small{\pm{1.5}}&77.6\small{\pm{1.7}}&80.1\small{\pm{0.9}} &74.5\newline
> \text{FragCL}(\alpha=1.0) &{70.9}\small{\pm{1.6}} & {76.2}\small{\pm{0.2}} & {64.2}\small{\pm{0.5}} & 61.9\small{\pm{0.9}} & {89.9}\small{\pm{1.2}} & {77.8}\small{\pm{0.6}} & {77.8}\small{\pm{0.5}} & {80.9}\small{\pm{1.0}} & 75.0 \newline
> \text{FragCL}(\alpha=2.0) &69.2\small{\pm{0.2}}&75.3\small{\pm{0.7}}&64.0\small{\pm{0.3}}&60.3\small{\pm{0.1}}&92.0\small{\pm{2.7}}&76.6\small{\pm{1.4}}&77.2\small{\pm{0.6}}&	79.6\small{\pm{0.5}}&74.3 \newline
> \hline
> \end{array}
>
>
>
> ------------
> **(Q2) Improvements in downstream prediction tasks are only modest.**
>
> Our gain in downstream prediction tasks are not marginal, given the prior arts. First of all, our method achieves the state-of-the-art performance for 7 out of 8 downstream tasks of MoleculeNet dataset, and all downstream tasks of ATOM3D dataset. Especially, our improvement is consistent over downstream tasks.
>
> ------------
> **(Q3) Torsion angle both as a regression problem and as a multiclass classification problem.**
>
> We clarify how our torsion angle reconstruction is formulated as a classification, not a regression. Specifically, our torsional angle reconstruction loss consists of (1) the binary classification problem for the rotation direction (i.e., clockwise vs. counter-clockwise) and (2) the multiclass classification problem for bins of torsional angles \{0 to $\pi/9$, $\pi/9$ to $2\pi/9$, …, $8\pi/9$ to $\pi$\}. We note that there is no redundancy between our objectives (i.e., (1) and (2) contains non-overlapping information to reconstruct torsional angles).
>
> ----------
> **(Q4) Single bonds are oftentimes highly important in reaction pathways.**
>
> Thank you for the opportunity to clarify this point. Single bonds can be involved in the reactions such as SN1 and SN2 reactions. However, our fragmentation scheme disconnects the molecule in half, which does not influence the groups capable of SN1 or SN2 reactions (e.g., a single halogen atom) in general. We clarified this in our revision (Section 3.1).

---

> ### Author Response · Authors · 2022-12-07
> **A Gentle Reminder**
>
> Dear Reviewer ov3H,
>
> Thank you very much again for your time and efforts in reviewing our paper.
>
> We kindly remind that we have only a week or so in the discussion period.
>
> We just wonder whether there is any further concern and hope to have a chance to respond before the discussion phase ends.
>
> Many thanks, Authors

---

### Official Review · Reviewer_NCgJ · 2022-10-24

**Confidence:** 5
**Correctness:** 1
**Technical Novelty And Significance:** 2
**Empirical Novelty And Significance:** 1
**Recommendation:** 3

**Clarity, Quality, Novelty And Reproducibility:**

- The method naming is confusing. FragCL only implies the contrastive learning, but actually there is an torsion angle part, which is not reflected in FragCL.
- Some references are mis-match. I would recommend authors double-check this.


**Strength And Weaknesses:**

Strength:
- The existing works on utilizing the domain knowledge to help molecule representation is either focusing on the 2D structure (like fragments) or general 3D and 2D structure. This paper introduces using the fragments from two views, which is interesting and promising.
- This paper further introduces using the torsion angle reconstruction, which can help the molecule representation intuitively.


Weakness:
- There exists information leakage between the SSL objective and backbone model.
  - The authors are using SchNet for the geometric backbone. It is an SE(3)-invariant model, using the pairwise distance information, and adding torsion angle to it has been expected to be helpful. This has been verified in GemNet[1] and SphereNet[2].
  - Meanwhile, there are more advanced SE(3)-invariant GNNs (like GemNet[1] and SphereNet[2]). They all explicitly use the torsion angles directly in the backbone models. Other SE(3)-equivariant GNNs are also preferred, like TFN [3], since the angle information is simultaneously modeled in the spherical harmonics basis.
  -  In other words, using the SchNet as backbone is not sufficient to support the effect of using torsion angle reconstruction. Now the information released in the experiments is that for geometric model with distance modeling only, adding torsion angle pretraining is beneficial. This is different from the main claim of the paper, because this may fail if the backbone models [1,2,3] explicitly/implicitly encodes the torsion angle.
  - For improvements, the authors should use more advanced backbone models with torsion angle modeling, either implicitly or explicitly. Then it can better verify the effectiveness of using torsion angle for pretraining.

- The experiments are not solid for the 3D downstream.
  - First, the 7 tasks are actually from QM9, with 12 tasks and 130K molecules.
  - So the pretraining has 50K molecules, and the downstream has 130K molecules. I don’t think this is a valid pretraining and finetuning setting.
  - The data preprocessing has been widely used on QM9 (SchNet, DimeNet, SphereNet) etc, but the authors are not applying that. So this result can be hardly compared with other existing works.
  - There are 12 tasks, I’m wondering why only 7 tasks are reported here?
  - Thus, combined all the 4 points above, I am not convinced by the validness of QM9 downstream results.

- Questions on 2D downstream. I found that authors are reporting results on 2D-FragCL, FragCL, and 3D InfoMax, and the other results are from GraphMVP paper, which is fine. But I noticed that the main benefits between the newly reported results and existing baselines are from Clintox, which is >0.1 ROC-AUC better. Other than this, the improvements are quite modes, within 0.01 ROC-AUC. Do the authors have any hypothesis for this? Could it be the hyper-parameter issue (seed, learning rate, etc)?

[1] Gasteiger, Johannes, Florian Becker, and Stephan Günnemann. "Gemnet: Universal directional graph neural networks for molecules." Advances in Neural Information Processing Systems 34 (2021): 6790-6802.

[2] Liu, Yi, et al. "Spherical message passing for 3d graph networks." arXiv preprint arXiv:2102.05013 (2021).

[3] Thomas, Nathaniel, et al. "Tensor field networks: Rotation-and translation-equivariant neural networks for 3d point clouds." arXiv preprint arXiv:1802.08219 (2018).


**Summary Of The Paper:**

This paper decomposes the 2D and 3D molecular graphs into fragments. The pretraining objective is composed of the fragment-based CL on 2D and 3D graphs, as well as a torsion angle reconstruction from 2D to 3D.


**Summary Of The Review:**

I think the main motivation of this paper is reasonable, but the experimental set-ups are not valid. The authors may double-check the SE(3)-invariant/equivariant modeling, and the QM9 dataset.

---

> ### Author Response · Authors · 2022-11-14
> **Response to Reviewer NCgJ (3/3)**
>
> **(Q3) Data preprocessing compared to prior works.**
>
> We simply followed the data preprocessing of the official ATOM3D benchmark paper [1]. To alleviate your concern, we re-trained the models with the suggested data preprocessing scheme. As expected, the results show that our method still outperforms the prior arts (see the above table in Q2). We here note that the referred papers (SchNet [2], DimeNet [3], and SphereNet [4]) focus on designing a neural network for 3D molecule point-cloud while our focus is self-supervised training of 2D encoder; Namely, our results are not directly comparable with the referred papers [2-4].
>
> [1] Townshend et al., “ATOM3D: Tasks On Molecules in Three Dimensions”, arXiv 2020
>
> [2] Schutt et al., “SchNet: A Continuous-filter Convolutional Neural Network for Modeling Quantum Interactions”, NeurIPS 2017
>
> [3] Gasteiger et  al., “Directional Message Passing for Molecular Graphs”, ICLR 2020
>
> [4] Liu et al., “Spherical Message Passing for 3D Molecular Graphs”, ICLR 2022
>
> -----------
> **(Q4) The choice of downstream tasks.**
>
> We first excluded ‘$U_{298}$’, ‘$H_{298}$’, and '$G_{298}$’ following the choice of our baseline (3D-InfoMax) paper [1]. We then additionally excluded ‘$\text{HOMO}$’ and ‘$\text{LUMO}$’ since ‘$\varepsilon_{gap}:=|\text{HOMO}-\text{LUMO}|$’ is more meaningful and mainly defines the molecular property [2]. Nevertheless, to address your concern, we here report results on all the tasks with ATOM3D pretraining and common preprocessing as suggested (see the above table in Q2). As expected, our framework still achieves state-of-the-art performance for all the downstream tasks.
>
> [1] Stark et al., “3D Infomax improves GNNs for Molecular Property Prediction”, ICML 2022
>
> [2] Smith, “Organic Chemistry”, McGraw Hill, 6th Ed
>
> ---------
> **(Q5) I noticed that the main benefits between the newly reported results and existing baselines are from Clintox, which is >0.1 ROC-AUC better. Other than this, the improvements are quite modes, within 0.01 ROC-AUC. Do the authors have any hypothesis for this? Could it be the hyper-parameter issue (seed, learning rate, etc)?**
>
> Not all of our newly reported baselines benefit from the Clintox task (e.g., JOAOv2, MGSSL, and MolCLR). Hence, we think the task benefits from our FragCL components as shown in our component analysis (see Table 9). Also, we would like to emphasize that our “consistent” improvements of FragCL are significant enough compared to the prior arts. For example, in Table 1, GraphMVP-C achieves the best ROC-AUC for Bace, Sider and BBBP but fails to generalize on Tox21 (5th out of 5 2D and 3D pretrained baselines). In contrast, FragCL achieves state-of-the-art performance in 7 out of 8 tasks without severe degradation on a specific task, verifying the significance of our method.
>
> ----------
> **(Q6) “FragCL” only implies contrastive learning. Torsion angle part is not reflected.**
>
> Thank you for the suggestion of our method name. However, we would like to maintain the name “FragCL” since we want to emphasize the fragment-based contrastive learning (e.g., positive/negative view construction) as our main focus. Also, we remark that our torsional angle reconstruction tasks also utilize the fragments of a molecule, supporting the rationale of our naming.
>
> ---------
> **(Q7) Reference mismatch.**
>
> Many thanks for the careful reading. We carefully revised our manuscript to remove the mismatches.

---

> > ### Comment · Reviewer_NCgJ · 2022-11-17
> > **Main Concerns Remain Unsolved**
> >
> > I appreciate the authors for the reply. However, my main concerns remain unsolved.
> >
> > 1. **The results in Sec G don't address the torsion-angle issue.** As in the initial comment, the torsion-angle is very sensitive to the geometry-related tasks, like QM9, and that's what I have been concerning about. Currently, Sec G is about 2D property prediction downstream, which is not my main concern. If the authors want to claim that the effectiveness of the proposed method is not caused by the torsion angle, then downstream tasks on molecule geometry are required.
> > 2. **The results on QM9 (Table 2) are still confusing.** I appreciate the authors adopting the data transformation on the related works. However, the results are still confusing. The backbone model is SchNet, and its supervised results appear in those related works; meanwhile, it should also match with the first row in Tabe 2, which is not the case for now. I suggest the authors double-check it.
> > 3. Based on the confusing results in Table 2, I cannot tell the validity of results on 3D downstream tasks and whether or not a small pretraining dataset makes sense. What I can tell is, there are quite some parallel works on pretraining and finetuning for 3D tasks, and they are all following the same pipeline of data preprocessing, i.e., the ones I shared last time. Currently, Table 2 seems to be an outlier to these works, so I cannot tell the effectiveness of FragCL. I strongly recommend the authors to fix this, otherwise it would be very hard for readers/followers to check this work along the research line.
> > 4. **Questions on the data preprocessing**. The authors confirmed that the previous data preprocessing could be the issue for confusing results in Table 2 (`We simply followed the data preprocessing of the official ATOM3D benchmark paper`), and now authors simply adopt the preprocessing steps as suggested in my previous comment. So what I'm curious is what exactly are the differences in QM9 preprocessing steps between Atom3D and other works? And is this related to question 2 above? Can authors carefully check this? Since now the results are still confusing, could it be other issues that lead to the mismatch?
> > 5. **Current objective function is wrong. Need to update in the future.** If the authors run pretraining in the future, I think authors can update the objective function in Eq 5. Currently Eq 5 is $\frac{\text{pos}}{\sum \text{neg}}$, which is not NT-Xnet. I know many graph SSL papers are wrong, simply because they follow a same graph SSL work (which is wrong). Simply `following` previous work is dangerous, because previous work can be problematic.
> >    - I didn't take this as a negative comment in the previous comment because many papers are wrong, and I know that you are just following their objective function.
> >    - But since that you can rerun the pretraining on SphereNet in one week, I strongly recommend you to use the right objective function and try downstream on QM9 in the next revision.
> >
> >
> >
> > Some suggestions to authors:
> > - Since now you are using the SphereNet, you can simply follow their code base, which is highly reproducible.
> > - You are only using the QM9 task in Atom3D, and QM9 is more well-known in the computation chemistry domain, so you should just mention QM9 in the paper. Currently, authors are saying words like `Atom3D downstream tasks`, but Atom3D has 7 more tasks that haven't been used in this work. So words like this (`Atom3D downstream tasks`) are inappropriate.

---

> > > ### Author Response · Authors · 2022-11-27
> > > **Response to Reviewer NCgJ (2/2)**
> > >
> > > **(Q5) Current objective function is wrong.**
> > >
> > > As we mentioned in the draft, we use a widely-used variant of NT-Xent which also theoretically maximizes the mutual information of positive views [1]. However, following your suggestion, we report the result with the original NT-Xent loss and SphereNet as 3D encoder (in pretraining). In the table below, the result from the suggested setup still outperforms the baselines.
> > >
> > > \begin{array}{l|cccccccccccc}
> > > \text{Methods} & \text{ZPVE} & \text{$\mu$} & \text{$\alpha$} & \text{$C_v$} & \text{LUMO} & \text{HOMO}& \text{$\epsilon_{gap}$} & \text{$R^2$} & \text{$U_0$} & \text{$U_{298}$} & \text{$H_{298}$} & \text{$G_{298}$}\newline
> > > \hline
> > > \text{-} & 49.7\small{\pm{8.7}}&0.428\small{\pm{0.002}}&0.666\small{\pm{0.060}}&0.255\small{\pm{0.008}}&84.8\small{\pm{0.7}}&85.6\small{\pm{1.2}}&124\small{\pm{1}}&28.8\small{\pm{0.9}}&74.9\small{\pm{9.5}}&68.3\small{\pm{11.2}}&72.0\small{\pm{10.6}}&71.2\small{\pm{2.9}} \newline
> > > \text{CP} &30.7\small{\pm{2.1}}&0.416\small{\pm{0.002}}&0.633\small{\pm{0.032}}&0.219\small{\pm{0.005}}&85.6\small{\pm{0.9}}&86.7\small{\pm{0.9}}&124\small{\pm{2}}&25.4\small{\pm{0.3}}&60.0\small{\pm{6.1}}&60.8\small{\pm{10.1}}&65.2\small{\pm{11.1}}&60.8\small{\pm{10.1}}\newline
> > > \text{GraphCL} & 27.2\small{\pm{0.5}}&0.419\small{\pm{0.005}}&0.589\small{\pm{0.039}}&0.225\small{\pm{0.002}}&85.0\small{\pm{0.5}}&85.3\small{\pm{0.3}}&121\small{\pm{1}}&25.1\small{\pm{0.4}}&57.2\small{\pm{4.2}}&58.9\small{\pm{3.9}}&55.1\small{\pm{2.7}}&59.9\small{\pm{4.5}} \newline
> > > \text{GraphMVP-C} &  \textbf{22.7}\small{\pm{1.8}}&0.423\small{\pm{0.001}}&0.521\small{\pm{0.038}}&\textbf{0.199}\small{\pm{0.008}}&85.2\small{\pm{0.6}}&86.5\small{\pm{0.3}}&124\small{\pm{1}}&\textbf{25.2}\small{\pm{0.1}}&\textbf{37.4}\small{\pm{2.0}}&\textbf{38.6}\small{\pm{3.6}}&\textbf{39.2}\small{\pm{4.0}}&43.6\small{\pm{2.2}} \newline
> > > \text{3D-Infomax} &\textbf{22.2}\small{\pm{1.6}}&0.412\small{\pm{0.004}}&\textbf{0.492} \small{\pm{0.006}}&\textbf{0.202}\small{\pm{0.004}}&84.9\small{\pm{1.0}}&\textbf{83.2}\small{\pm{0.9}}&121\small{\pm{0}}&\textbf{24.8}\small{\pm{0.3}}&{40.3}\small{\pm{1.3}}&\textbf{39.7}\small{\pm{1.2}}&\textbf{38.7}\small{\pm{2.8}}&\textbf{38.0}\small{\pm{1.3}}\newline
> > > \hline
> > > \textbf{FragCL}& \textbf{23.4}\small{\pm{1.2}}&\textbf{0.409}\small{\pm{0.001}}&\textbf{0.488}\small{\pm{0.023}}&\textbf{0.201}\small{\pm{0.007}}&\textbf{80.9}\small{\pm{1.3}}&\textbf{83.0}\small{\pm{0.6}}&\textbf{118}\small{\pm{1}}&\textbf{24.4}\small{\pm{0.8}}&\textbf{39.1}\small{\pm{3.0}}&\textbf{40.0}\small{\pm{3.6}}&\textbf{40.9}\small{\pm{2.9}}&\textbf{37.7}\small{\pm{2.2}}\newline
> > > \hline
> > > \textbf{FragCL (NT-Xent,SphereNet)}& {23.7}\small{\pm{1.1}}&{0.411}\small{\pm{0.002}}&\textbf{0.486}\small{\pm{0.028}}&\textbf{0.201}\small{\pm{0.009}}&\textbf{80.5}\small{\pm{0.6}}&\textbf{82.3}\small{\pm{1.0}}&\textbf{118}\small{\pm{1}}&\textbf{24.6}\small{\pm{0.1}}&\textbf{39.1}\small{\pm{2.3}}&\textbf{39.1}\small{\pm{2.3}}&\textbf{41.4}\small{\pm{2.0}}&\textbf{38.6}\small{\pm{1.8}}\\
> > > \end{array}
> > >
> > > [1] You et al., “Graph Contrastive Learning with Augmentations”, NeurIPS 2020
> > >
> > >
> > > ------------
> > > **(Q6) Editorial comment**
> > >
> > > Thanks for the comment. The suggestion will be carefully incorporated into the final manuscript.

---

> > > ### Author Response · Authors · 2022-11-27
> > > **Response to Reviewer NCgJ (1/2)**
> > >
> > > Dear Reviewer NCgJ
> > >
> > > Thank you for your comments.
> > >
> > > We want to clarify how the works of 3D GNNs [1,2,3] are not comparable to our setup (Q1-Q4). Following prior works [4,5], we utilize **only the (pretrained) 2D GNNs** for various downstream tasks (i.e., MoleculeNet and QM9) **without 3D information**. Since [1,2,3] rely on supervised training on 3D information, they are not directly comparable to our work.
> > >
> > > [1] Gasteiger, Johannes, Florian Becker, and Stephan Günnemann. "Gemnet: Universal directional graph neural networks for molecules." Advances in Neural Information Processing Systems 34 (2021): 6790-6802.
> > >
> > > [2] Liu, Yi, et al. "Spherical message passing for 3d graph networks." arXiv preprint arXiv:2102.05013 (2021).
> > >
> > > [3] Thomas, Nathaniel, et al. "Tensor field networks: Rotation-and translation-equivariant neural networks for 3d point clouds." arXiv preprint arXiv:1802.08219 (2018).
> > >
> > > [4] Liu et al., “Pre-training Molecular Graph Representation with 3D Geometry”, ICLR 2022
> > >
> > > [5] Stark et al., “3D Infomax improves GNNs for Molecular Property Prediction”, ICML 2022
> > >
> > >
> > > -------
> > > **(Q1) Sec G does not address the torsion-angle issue. Downstream tasks on molecule geometry are required.**
> > >
> > > Sec G does address the torsion-angle issue, i.e., the results of Sec G confirms the effectiveness of the torsional angle reconstruction task. We validate the effectiveness of the torsional angle reconstruction task by using SphereNet as 3D encoder, as you suggested. We note that  our work focuses on fine-tuning 2D GNN on both MoleculeNet and QM9 experiments (i.e., downstream tasks on 2D molecular graphs). “Downstream tasks on 3D molecule geometry” is beyond our scope.
> > >
> > > ---------
> > > **(Q2) The results on Table 2 are still confusing.**
> > >
> > > The first row in Table 2 is based on GIN (2D GNN), not SchNet (3D GNN). Our result is comparable with other 2D GNN results (see Table 1 first column of [1]): e.g., our GIN and PNA (2D GNN) of [1] achieve 84.8, 85.6, 124 and 82.10, 85.72, 123.08 on HOMO, LUMO, and $\varepsilon$, respectively.
> > >
> > > [1] Stark et al., “3D Infomax improves GNNs for Molecular Property Prediction”, ICML 2022
> > >
> > > ----------
> > > **(Q3) Table 2 seems to be an outlier to prior works.**
> > >
> > > Table 2 is not an outlier to prior works. As mentioned in Q2, the results of Table 2 match with the table from our compatible baseline, 3D-Infomax [1]. This confirms the validity of our experimental setting. The works you mentioned are about 3D GNN, which is not compatible with our work (about the fine-tuning of 2D GNN).
> > >
> > > ----------
> > > **(Q4) Questions on the data preprocessing.**
> > >
> > > The main difference of data preprocessing was the unit of the data. For example, the unit of $\varepsilon$ is eV in ATOM3D, while meV in the mentioned works. Also, ATOM3D utilizes the normalization of training labels by mean and standard deviation. Such experimental choices are not related to the Q2 above. Also, as mentioned in Q2, our results do align with the prior work.

---

> ### Author Response · Authors · 2022-11-14
> **Response to Reviewer NCgJ (2/3)**
>
> **(Q2) Validity of pretraining and fine-tuning dataset.**
>
> Thanks for the advice of the experimental setting. We still think our transfer learning (GEOM → ATOM3D) setting is meaningful, irrespective of the numbers of training samples used for pretraining and fine-tuning. Despite the difference between the datasets and the small number of pretraining molecules, our method significantly improves the performance in 3D downstream tasks as shown in Table 2 of the original draft.
>
> Nevertheless, to alleviate your concern, we pretrain both our model and baselines on QM9 and then fine-tune them on the 3D downstream tasks (with the suggested data preprocessing of [1,2,3]). Results show that our method achieves the state-of-the-art performance for all downstream tasks. We updated Table 2 in the manuscript following your suggestions (e.g., pretraining dataset, data processing, and the number of downstream tasks).
>
> \begin{array}{l|cccccccccccc}
> \text{Methods} & \text{ZPVE} & \text{$\mu$} & \text{$\alpha$} & \text{$C_v$} & \text{LUMO} & \text{HOMO}& \text{$\epsilon_{gap}$} & \text{$R^2$} & \text{$U_0$} & \text{$U_{298}$} & \text{$H_{298}$} & \text{$G_{298}$}\newline
> \hline
> \text{-} & 49.7\small{\pm{8.7}}&0.428\small{\pm{0.002}}&0.666\small{\pm{0.060}}&0.255\small{\pm{0.008}}&84.8\small{\pm{0.7}}&85.6\small{\pm{1.2}}&124\small{\pm{1}}&28.8\small{\pm{0.9}}&74.9\small{\pm{9.5}}&68.3\small{\pm{11.2}}&72.0\small{\pm{10.6}}&71.2\small{\pm{2.9}} \newline
> \text{CP} &30.7\small{\pm{2.1}}&0.416\small{\pm{0.002}}&0.633\small{\pm{0.032}}&0.219\small{\pm{0.005}}&85.6\small{\pm{0.9}}&86.7\small{\pm{0.9}}&124\small{\pm{2}}&25.4\small{\pm{0.3}}&60.0\small{\pm{6.1}}&60.8\small{\pm{10.1}}&65.2\small{\pm{11.1}}&60.8\small{\pm{10.1}}\newline
> \text{GraphCL} & 27.2\small{\pm{0.5}}&0.419\small{\pm{0.005}}&0.589\small{\pm{0.039}}&0.225\small{\pm{0.002}}&85.0\small{\pm{0.5}}&85.3\small{\pm{0.3}}&121\small{\pm{1}}&25.1\small{\pm{0.4}}&57.2\small{\pm{4.2}}&58.9\small{\pm{3.9}}&55.1\small{\pm{2.7}}&59.9\small{\pm{4.5}} \newline
> \text{GraphMVP-C} &  \textbf{22.7}\small{\pm{1.8}}&0.423\small{\pm{0.001}}&0.521\small{\pm{0.038}}&\textbf{0.199}\small{\pm{0.008}}&85.2\small{\pm{0.6}}&86.5\small{\pm{0.3}}&124\small{\pm{1}}&\textbf{25.2}\small{\pm{0.1}}&\textbf{37.4}\small{\pm{2.0}}&\textbf{38.6}\small{\pm{3.6}}&\textbf{39.2}\small{\pm{4.0}}&43.6\small{\pm{2.2}} \newline
> \text{3D-Infomax} &\textbf{22.2}\small{\pm{1.6}}&0.412\small{\pm{0.004}}&\textbf{0.492} \small{\pm{0.006}}&\textbf{0.202}\small{\pm{0.004}}&84.9\small{\pm{1.0}}&\textbf{83.2}\small{\pm{0.9}}&121\small{\pm{0}}&\textbf{24.8}\small{\pm{0.3}}&{40.3}\small{\pm{1.3}}&\textbf{39.7}\small{\pm{1.2}}&\textbf{38.7}\small{\pm{2.8}}&\textbf{38.0}\small{\pm{1.3}}\newline
> \hline
> \textbf{FragCL (Ours)}& \textbf{23.4}\small{\pm{1.2}}&\textbf{0.409}\small{\pm{0.001}}&\textbf{0.488}\small{\pm{0.023}}&\textbf{0.201}\small{\pm{0.007}}&\textbf{80.9}\small{\pm{1.3}}&\textbf{83.0}\small{\pm{0.6}}&\textbf{118}\small{\pm{1}}&\textbf{24.4}\small{\pm{0.8}}&\textbf{39.1}\small{\pm{3.0}}&\textbf{40.0}\small{\pm{3.6}}&\textbf{40.9}\small{\pm{2.9}}&\textbf{37.7}\small{\pm{2.2}}\\
> \end{array}
>
> [1] Schutt et al., “SchNet: A Continuous-filter Convolutional Neural Network for Modeling Quantum Interactions”, NeurIPS 2017
>
> [2] Gasteiger et  al., “Directional Message Passing for Molecular Graphs”, ICLR 2020
>
> [3] Liu et al., “Spherical Message Passing for 3D Molecular Graphs”, ICLR 2022

---

> ### Author Response · Authors · 2022-11-14
> **Response to Reviewer NCgJ (1/3)**
>
> Dear reviewer NCgJ,
>
> We sincerely appreciate your efforts in reviewing our manuscript. We respond to each comment in the following content. We carefully incorporated the discussions into the final manuscript.
>
> --------
> **(Q1) Backbone model as torsional-angle aware 3D architectures.**
>
> First, our architectural choice is for the fair comparison with GraphMVP (one of our main baselines). Also, our torsional angle reconstruction task is still meaningful even with torsion-angle-aware 3D encoders (e.g., SphereNet [1]) because the task enables our 2D encoder to explicitly learn node-level 3D information while the contrastive objective between 2D and 3D encoder (i.e., Eq. (8) in our paper) only shares graph-level information. To support this and alleviate your concern, we additionally conduct experiments with SphereNet. In the table below, the torsion reconstruction task still improves the performance (74.4 -> 75.1), demonstrating the effectiveness of torsion reconstruction task.
>
> \begin{array}{l|cccccccc|c}
> \hline
> \text{SphereNet} & \text{BBBP} & \text{Tox21} & \text{ToxCast} & \text{Sider} & \text{Clintox} & \text{MUV} & \text{HIV} & \text{Bace} & \text{Avg.}\newline
> \hline
> \mathcal{L}_\mathtt{con} & 69.8\small{\pm{1.3}} & 75.6\small{\pm{0.3}} & 64.9\small{\pm{0.6}} & 62.0\small{\pm{1.1}} & 89.9\small{\pm{1.7}} & 75.9\small{\pm{0.4}} & 76.4\small{\pm{0.3}} & 80.3\small{\pm{0.9}} & 74.4 \newline
> \mathcal{L}_\mathtt{FragCL} & 71.5\small{\pm{1.4}} & 75.7\small{\pm{0.7}} & 65.7\small{\pm{0.3}} & 61.3\small{\pm{1.0}} & 91.4\small{\pm{1.4}} & 78.1\small{\pm{2.2}} & 76.1\small{\pm{1.6}} & 80.7\small{\pm{0.6}} & 75.1\newline
> \hline
> \end{array}
>
> [1] Liu et al., “Spherical Message Passing for 3D Molecular Graphs”, ICLR 2022

---

> ### Author Response · Authors · 2022-12-07
> **A Gentle Reminder**
>
> Dear Reviewer NCgJ,
>
> Thank you very much again for your time and efforts in reviewing our paper.
>
> We kindly remind that we have only a week or so in the discussion period.
>
> We just wonder whether there is any further concern and hope to have a chance to respond before the discussion phase ends.
>
> Many thanks, Authors

---

### Official Review · Reviewer_yCZe · 2022-10-25

**Confidence:** 4
**Correctness:** 3
**Technical Novelty And Significance:** 3
**Empirical Novelty And Significance:** 3
**Recommendation:** 8

**Clarity, Quality, Novelty And Reproducibility:**

Figure 1 paragraph description is a bit confusing. It is unclear upon the first read that the words in parenthesis correspond to one another (incomplete / negative).

What is the ratio of hard to soft negatives in the objective? How is that determined or optimized?

**Strength And Weaknesses:**

I appreciated the number of baselines presented, and I think the proposed model results are rather good.

I really like that the objective is straightforward and pragmatic. The approach is straightforward and more generally applicable. Since the model inputs and objectives are quite interpetable, I think Section E (Table 9) is useful as an ablation study.

Questions:

What is the distribution of 3D bond angles used in the objective? Was only a single bond angle used, and where was this generated? Would it be possible to learn using the energetically favorable distribution of bond angles?

The loss for the torsion angle is interesting and pragmatic. However, by using a binned loss for the torsion angles, it is not radially symmetric, and may not be ideal. Can the angle loss be parameterized with a von Mises distribution?

**Summary Of The Paper:**

The authors develop a graph neural network model that breaks a module up into fragments over non-ring single bonds, and trains a both a contrastive bag-of-fragments objective as well as prediction of the torsion angle of the respective broken bond.

**Summary Of The Review:**

I find the approach novel, pragmatic, and of general interest. I also find the work well written and easy to understand.

---

> ### Author Response · Authors · 2022-11-14
> **Response to Reviewer yCZe**
>
> Dear reviewer yCZe,
>
> We sincerely appreciate your efforts in reviewing our manuscript. We respond to each comment in the following content. We carefully incorporated the discussions into the final manuscript.
>
> -------------
> **(Q1) What is the distribution of 3D bond angles used in the objective? Was only a single bond angle used, and where was this generated?**
>
> As you mentioned, we only consider the torsional angle around the single bond disconnected in the fragmented step. We here provide the distribution of torsional angles in the below table. Following the chemical knowledge [1], peaks occur at anti position (around 180 degree) and gauche position (around 60 and 300 degree). Additionally, due to the resonance-related single bonds, there exists another peak around 0 degree.
>
> \begin{array}{l|cccccccccccccccccc}
> \hline
> \text{Degree} & \text{0-20} & \text{20-40} & \text{40-60} & \text{60-80} & \text{80-100} & \text{100-120} & \text{120-140} & \text{140-160} & \text{160-180} & \text{180-200} & \text{200-220} & \text{220-240} & \text{240-260} & \text{260-280} & \text{280-300} & \text{300-320} & \text{320-340} & \text{340-360} \newline
> \hline
> \text{(\\%)} & 8.2 & 3.5 & 5.1 & 7.2 & 5.2 & 3.5 & 3.0 & 4.0 & 10.6 & 10.4 & 3.9 & 3.2 & 3.3 & 5.2 & 7.0 & 5.4 & 3.5 & 8.0 \newline
> \hline
> \end{array}
>
> [1] Smith, “Organic Chemistry”, McGraw Hill, 6th Ed
>
> ------------
> **(Q2) Would it be possible to learn using the energetically favorable distribution of bond angles?**
>
> Yes, it is. In fact, our method already utilizes the energetically favorable distribution of bond angles, by learning the distribution of bond angles which is related to the energy via Boltzman distribution, i.e., the probability of a bond angle is proportional to $e^{-\varepsilon/kT}$ where $\varepsilon$ is the energy of a given bond angle.
>
> ------------
> **(Q3) Can the loss be parametrized with a von Mises distribution?**
>
> Thank you for the constructive suggestion. The loss can be parametrized with a von Mises distribution, but we think our binned objective is more reasonable than a radially symmetric distribution (e.g., a von Mises distribution) since a torsional angle distribution given anchor atoms of a molecule is asymmetric and multimodal (e.g., see
> [here](https://forums.studentdoctor.net/attachments/1614288167642-png.331233)). However, it would be an interesting future work to parametrize the torsional angle loss via more sophisticated parametrization, e.g., a mixture of von Mises distributions.
>
> ----------
> **(Q4) Comment about the description of Figure 1.**
>
> Thank you for your suggestion to improve our manuscript. Following your suggestion, we updated the description for clarity.
>
> -----------
> **(Q5) What is the ratio of hard to soft negatives in the objective? How is that determined or optimized?**
>
> The ratio is determined by (batch size-1) : 2 for soft and hard negative samples. We remark how we use (a) complete bags of fragments of other molecules in the batch as soft negatives and (b) incomplete bags of fragments of its own molecule as hard negative samples.

---

> ### Author Response · Authors · 2022-12-07
> **A Gentle Reminder**
>
> Dear Reviewer yCZe,
>
> Thank you very much again for your time and efforts in reviewing our paper.
>
> We kindly remind that we have only a week or so in the discussion period.
>
> We just wonder whether there is any further concern and hope to have a chance to respond before the discussion phase ends.
>
> Many thanks, Authors

---

### Author Response · Authors · 2022-11-14
**General Response**

Dear reviewers and AC,

We sincerely thank all the reviewers and AC for your enormous effort and time spent reviewing our manuscript.

We also appreciate all the positive comments from reviewers: e.g., the method being interesting (all reviewers) and novel (ov3H,EnEZ), decent empirical improvements (yCZe,ov3H), intuitive objectives (yCZe,NcgJ,EnEZ), and well-written manuscript (yCZe,EnEZ). We believe that our paper proposes a simple yet effective contrastive learning framework comprising a chemical knowledge: a molecule can be viewed as a “bag of fragments” of meaningful substructures.

Thanks to the valuable comments on our manuscript, we have carefully revised and improved the manuscript in response to your concerns. The following list consists of newly added discussions and experimental results.

- Discussions and experiments about the preservation of chemically informative structures under fragmentation scheme (Section 4.3, Appendix F)

- Extended evaluation of downstream tasks on ATOM3D dataset (Table 2)

- Extended ablation study about the negative view construction and fragmentation strategy (Table 3)

- Clarified description of experimental settings (Section 4.1, Appendix A)

- Ablation study on the hyperparameter (Appendix E, Table 10)

- Discussions and experiments about the torsion-aware architecture as 3D encoder (Section 4.3, Appendix G)

We temporarily highlighted the revised contents in "blue” for your convenience to check.  We believe that FragCL can be a useful addition to the ICLR and molecule representation learning community. Also, we hope that the above updates clarify the effectiveness of our method.

Thank you very much!

Best,\
Authors.

---

### Decision · Program_Chairs · 2023-01-20

**Decision:**

Reject

**Justification For Why Not Higher Score:**

Three of the reviewers had concerns regarding the strength of the experiments; one had severe concerns about the validity of the experimental setting and baselines. While the authors did a good job of responding to these concerns, they were not sufficient to cause the reviewers to raise their scores.

**Justification For Why Not Lower Score:**

N/A

**Metareview: Summary, Strengths And Weaknesses:**

This paper introduces a novel contrastive learning approach for learning unsupervised molecule representations, based off of an algorithm for splitting molecules into fragments. Reviewers praised the intuitive nature of the objectives, and the writing of the manuscript.

However reviewers were more mixed in their views when it came to evaluating the empirical performance. Some reviewers felt that the performance gains were relatively modest given the additional complexity; others had concerns regarding the experimental settings and the relation to the baselines.

The authors did a good job of responding to these points in the rebuttal and the updated paper. However, none of the three lean-reject reviews increased their score from the comments or discussion. That said I believe the updated manuscript, with the revised experimental details, is in a good position to be accepted when re-submitted at an upcoming conference.